# Assimilation of wide-swath altimetry water elevation anomalies to correct large-scale river routing model parameters

Charlotte Marie Emery[1,2], Sylvain Biancamaria[1], Aaron Boone[3], Sophie Ricci[4], Mélanie C. Rochoux[4], Vanessa Pedinotti[5], and Cédric H. David[2]

[1]LEGOS, 16 Avenue Edouard Belin, 31400 Toulouse, France.
[2]Jet Propulsion Laboratory, California Institute of Technology, Pasadena, CA, USA
[3]CNRM-GAME, Meteo-France, 42 Avenue Gaspard Coriolis, 31000 Toulouse, France
[4]CECI, Université de Toulouse, CERFACS, CNRS, 42 Avenue Gaspard Coriolis, 31057 Toulouse cedex 1, France
[5]Magellium, 1 Rue Ariane, 31520 Ramonville-Saint-Agne, France

**Correspondence:** Charlotte Marie Emery (charlotte.emery@jpl.nasa.com)

**Abstract.** Land surface models combined with river routing models are widely used to study the continental part of the water cycle. They give global estimates of water flows and storages but they are not without non-negligible uncertainties among which inexact input parameters play a significant part. The incoming Surface Water and Ocean Topography (SWOT) satellite mission, with a launch scheduled for 2021 and with a required lifetime of at least three years, will be dedicated to the measuring of water surface elevations, widths and surface slopes of rivers wider than 100 meters, at a global scale. SWOT will provide a significant amount of new observations for river hydrology and maybe combined, through data assimilation, with global-scale models in order to correct their input parameters and reduce their associated uncertainty. Comparing simulated water depths with measured water surface elevations remains however a challenge and can introduce within the system large bias. A promising alternative for assimilating water surface elevations consists of assimilating water surface elevation anomalies which do not depend on a reference surface. The objective of this study is to present a data assimilation platform based on the asynchronous ensemble Kalman filter (AEnKF) that can assimilate synthetic SWOT observations of water depths and water elevation anomalies to correct the input parameters of a large scale hydrologic model over a 21-day time window. The study is applied to the ISBA-CTRIP model over the Amazon basin and focuses on correcting the spatial distribution of the river Manning coefficients. The data assimilation algorithm, tested through a set of Observing System Simulation Experiments (OSSE), is able to retrieve the true value of the Manning coefficients within one assimilation cycle much of the time (basin-averaged Manning coefficients RMSEn is reduced from 33% to [1%-10%] after one assimilation cycle) and shows promising perspectives with assimilating water anomalies (basin-averaged Manning coefficients RMSEn is reduced from 33% to [1%-2%] when assimilating water surface elevation anomalies over one year) which allows us to overcome the issue of unknown bathymetry.

# 1 Introduction

Global Hydrological Models (GHM) are extensively exploited to study the continental component of the global water cycle (Doll et al., 2015; Sood and Smakhtin, 2015). Such models have been extensively developed over the past two decades in order to quantify freshwater flows and storage changes over continental surfaces (Bierkens, 2015). They are based on the coupling
of a Land Surface Model (LSM) with a River Routing Model (RRM). As an example, the ISBA-CTRIP (Decharme et al., 2019) hydrologic model results from the coupling of the ISBA LSM (Noilhan and Planton, 1989) and the TRIP RRM (Oki and Sud, 1998). LSMs simulate the energy and water balance at the soil-atmosphere-vegetation interface while RRMs emulate the lateral transfer of freshwater toward the continent-ocean interface. The current study focuses on the river component of the terrestrial water cycle simulated by RRM.

GHMs give a global view of the state of the water flow and storage at model spatial and temporal resolutions. Nonetheless, they suffer from multiple sources of uncertainties which are related to the model structure, the external forcing and the input parameters (Liu and Gupta, 2007; Renard et al., 2010). Model structure uncertainties initially arose from a lack of knowledge in the hydrologic processes or from simplifying assumptions made to limit simulation computational cost. Still, with the increase of computational power, models are more and more complex (Liu and Gupta, 2007; Melsen et al., 2016): they run at finer
spatial resolution, they include new physical processes and use an increasing number of fully distributed forcing and parameter datasets (Liu et al., 2012). This has led to an increase in the number of model input parameters and potentially inflates the model uncertainty in those parameters. Input parameters express the spatial and/or temporal properties of the system. The spatial scale of parameters measurable on the field may differ from the model scale, while other conceptual parameters are not directly observable and measurable on the field (Moradkhani et al., 2005; Melsen et al., 2016) and are inferred using
geomorphological empirical formula and/or indirect methods such as calibration (Gupta et al., 1998; Beven, 2012).

Another way in which to study the terrestrial water cycle is to use direct observations of the system. Most parts of the terrestrial water cycle are currently observed and measured from in situ or remote techniques (Sanoo et al., 2011; Vinukollu et al., 2011; Rodell et al., 2015). For the observations of rivers, in situ techniques measure river water elevations at gauge stations. In situ measurements are commonly very accurate and also frequent (i.e. sub-daily) but their main limitation is their spatially
sparse sampling and their decreasing number over recent decades at a global scale (International Association of Hydrological Sciences Ad Hoc Group on Global Water Sets et al., 2001). Coincidentally, remotely-sensed data provided by satellite missions have increased quite significantly since the 90's and deliver effective river observations. The most common instrument operating to assess river water levels remains the nadir altimeter. Nadir altimetry gives localized water elevation measurements along the satellite ground track. Initially, altimeters were designed to monitor ocean topography but their application has broadened
to the observation of lakes (Cretaux et al., 2009), floodplains (Birkett et al., 2002) and later on, rivers (Silva et al., 2010). Yet, their main limitation remains their limited spatial and temporal samplings: generally several days between two consecutive measurements at a limited number of locations. Besides, over continental surfaces, the signal is not always retrievable. Current river observations therefore provide a more accurate view of the river system than models but they are quite limited by their sparse availability in space and time.

The incoming Surface Water and Ocean Topography (SWOT) mission, jointly developed by NASA, CNES, CSA and UKSA and scheduled for a launch in 2021, will be dedicated to the observation of continental free surface water with a better spatial and temporal coverage than the current nadir missions (such as EnviSat, the JASON series or also Sentinel-3A/B). SWOT main payload called KaRIn, for Ka-band Radar INterferometer (Fj´ortoft et al., 2014), will observe surfaces under two swaths of 50 km each separated by a nadir-gap of 20 km and will have a near-global coverage. For hydrology, SWOT will observe rivers wider than 100 meters as well as lakes and wetlands larger than $250 \times 250$ m$^2$ within the latitudes 78° South and 78° North and with a revisit time of 21 days. SWOT will provide two-dimensional images of water surface elevations with a vertical accuracy of 10 cm when averaged over 1 km$^2$ of water area. Along with water surface elevation measurements in rivers, SWOT will also provide observations of river width, surface slope and estimates of discharge based on SWOT observations. SWOT will provide a significant amount of new data for surface hydrology. It will give an ensemble of constraints that will allow a better depiction of surface water in hydrological models. This new data could be combined or integrated into global-scale hydrological models in order to correct them and improve their performances and forecasting capabilities.

Data assimilation techniques are a set of mathematical methods which combine a physical model and related external measurements taking their relative uncertainties into account. Data assimilation aims at improving the model ability to forecast and/or emulate the physical system's evolution. For this purpose, data assimilation methods are built to correct either the model's outputs (state estimation) or the model's input parameters (parameter estimation or PE), sometimes both simultaneously. Data assimilation for state estimation has been widely applied in meteorology and oceanography, and is more and more developed for large-scale terrestrial hydrology (Clark et al., 2008; Michailovsky et al., 2013; Paiva et al., 2013; Emery et al., 2018). Data assimilation for PE in hydrology has been initially developed as a dynamic alternative to model calibration (Montzka et al., 2011; Panzeri et al., 2013; Ruiz et al., 2013; Shi et al., 2015). In most models, parameters are assumed to be constant in time whereas, in reality, they may vary seasonally or under evolving climate and/or anthropogenic conditions. Sequential data assimilation can therefore help track model parameters variations in time (Kurtz et al., 2012; Deng et al., 2016; Pathiraja et al., 2016). PE is also used to retrieve conceptual parameters of hydrologic models such as friction coefficients (Pedinotti et al., 2014; Oubanas et al., 2018; Hafliger et al., 2019) or residence time of quick- and slow-flow reservoirs and partition of runoff excess (Vrugt et al., 2012; Pathiraja et al., 2016) which can not be directly measured.

Before launch, in the preparatory phase, Observing System Simulation Experiments (OSSE) can be performed in orderto assess the benefits ofassimilating SWOT data into a hydrological model and to evaluate the most adapted methodologies to assimilate this data into models. Several studies assimilating synthetic and/or simplified SWOT like data have been published so as to evaluate the correction of river model state namely river depth (Andreadis et al., 2007; Biancamaria et al., 2011), river storages (Munier et al., 2015) and river discharges (Andreadis and Schumann, 2014) at various scales. But also, several studies focused on the possibility of using SWOT data to retrieve critical river parameters such as river bathymetry (Durand et al., 2008; Yoon et al., 2012; Mersel et al., 2013) and/or riverbed roughness/friction coefficient (Pedinotti et al., 2014; Oubanas et al., 2018; Hafliger et al., 2019). Indeed, SWOT is a scientific mission with a three year nominal lifetime. Therefore, SWOT observations will help to better calibrate hydrological models and to improve their performances even over time periods beyond its lifetime. Moreover, other studies using real remote-sensing data have also been published and give insight into the challenges related

to the assimilation of space-borne products such as Michailovsky et al. (2013); Michailovsky and Bauer-Gottwein (2014) and Emery et al. (2018) which assimilate nadir radar altimetry data.

In the present study, a data assimilation framework is used to correct input parameters of the large-scale ISBA-CTRIP model. More specifically, synthetic SWOT observations of water surface depths and anomalies are assimilated in order to correct the spatially-distributed riverbed friction coefficients (or Manning coefficients). As SWOT will not directly measure water depths (it provides water elevation and the bathymetry is required to derive water depth), the purpose of this study is to evaluate the possibility of assimilating water elevation anomalies to correct the model's parameters and to assess how the assimilation performances are impacted, compared to the direct assimilation of water depths. Assimilating water elevation anomalies is done to overcome a potential lack of bathymetry data.

This study is presented as a complementary study to that of Emery et al. (2018) which is dedicated to the state estimation (river storage and discharge) of the same ISBA-CTRIP model, using real satellite-based discharge products. The choice of the roughness coefficient as control variable was made following the results from the ISBA-CTRIP sensitivity analysis in Emery et al. (2016). In this preliminary study, the sensitivity of the simulated water depths and also anomalies to several river input parameters (such as riverbed width, depth, slope and also friction coefficient) was evaluated. The results showed that the highest sensitivity was in the Manning coefficient.

This study is, furthermore, also built on the conclusions from the work of Pedinotti et al. (2014). In our study, an Ensemble Kalman Filter (EnKF) is used (instead of the Extended Kalman Filter in Pedinotti et al. (2014)) to better account for the nonlinearities of the system and to better estimate the model errors. Also, Pedinotti et al. (2014) chose to update the Manning coefficients distribution at the grid cell scale and the question of equifinality arose (Beven and Freer, 2001) in their results. For the current study, it was decided to update the Manning coefficient distribution not at the grid-cell resolution, but at a coarser zonal resolution, by applying multiplying correcting factors uniformly over each zone, identical to the one used in Emery et al. (2016). Finally, Pedinotti et al. (2014) used an assimilation window of 2 days. This configuration resulted in updated Manning coefficient time series displaying "unrealistic jumps" with a frequency of about 20 days associated to the orbit repeat cycle (longer than the 2-day window). To avoid this phenomenon, the present study uses an assimilation window of 21 days corresponding to the current SWOT orbit repeat cycle.

Section 2 will first give a description of the ISBA-CTRIP model used for this study. Section 3 will present the particular data assimilation method developed for this study and finally, after presenting the assimilation strategy in Section 4, Sections 5 and 6 will give the data assimilation results.

## 2 Model

### 2.1 The ISBA-CTRIP large-scale hydrological model

The ISBA model (Noilhan and Planton, 1989) is a LSM defined at global-scale on a $0.5° \times 0.5°$ regular mesh grid that establishes the energy and water budget over continental surfaces. This study operates the ISBA-3L version based on a three-layer soil (Boone et al., 1999). The budget equations are solved separately on eachgrid cell. Still, larger-scale spatial patterns in

the radiative and precipitation forcing, the soil composition and the vegetation cover ensure spatial correlations between those cells. (for more details, see Decharme et al., 2012, 2019). In particular, ISBA gives a diagnostic of the surface runoff ($Q_{\text{ISBA,sur}}$) and the gravitational drainage ($Q_{\text{ISBA,sub}}$, , i.e. water percolating to the deep layers of the soil) later used as forcing inputs for the RRM denoted CTRIP.

The CTRIP model (Decharme et al., 2010, 2012, 2019), is defined on the same mesh grid as ISBA and follows a river network to laterally transfer water from one cell to another, down to the interface with the ocean (Oki and Sud, 1998). The study is based on the CTRIP version from Decharme et al. (2012) with three reservoirs, as illustrated in Figure 1a. The water mass [kg] stored in a groundwater reservoir $G$ and a floodplain reservoir $F$ interacts with the water mass in the surface reservoir $S$ representing the river. Only the surface reservoir $S$ is related to the river network and fills with the surface runoff $Q_{\text{ISBA,sur}}$,

the outflow from upstream cells and the delayed drainage $Q_{\text{ISBA,sub}}$ by means of the groundwater reservoir. Occasionally, when the amount of water in the river exceeds a given threshold (defined by the water level in the reservoir), the river spills into the floodplains.

## 2.2   CTRIP parameters

Within a $0.5° \times 0.5°$ cell, the surface reservoir is a unique river channel that may gather multiple real river branches. Its

rectangular cross-section is described by its slope $s$ [-], its width $W$ [m], its bankful depth $H_c$ [m], its length $L$ [m] and finally a Manning or friction coefficient $N$ [s m$^{-1/3}$] that assesses the reach resistance at the bottom of the river.

    Each cell's elevation is deduced from the STN-30p Digital Elevation Model (http://daac.ornl.gov/ISLSCP_II/islscpii.shtml). These elevations are then compared to determine the riverbed slope $s$. Global empirical geomorphologic relationships are used to define the river width $W$ and bankful depth $H_c$. The arc length between grid cell centers, inflated by a meandering factor $\mu$,

results in the river reach length $L$. More details on these parameters can be found in Oki and Sud (1998) and Decharme et al. (2012).

    The Manning coefficient $N$ is generally more complicated to estimate. Following Maidment (1993), it should take values between 0.025 and 0.03 for natural streams and values between 0.075 and 0.1 for smaller and mountainous tributaries and also floodplains. Global studies can apply either a constant (Beighley et al., 2009; Biancamaria et al., 2009) or a spatially-

distributed (Decharme et al., 2012) Manning coefficient. However, it is ordinarily accepted that this parameter should vary in space and even in time across the river catchment. Consequently, CTRIP uses a spatially-distributed Manning coefficient based on a simple linear relationship between the relative stream size in the current cell, denoted $SO$, and the size at the river mouth and the source cells, so that:

$$N = N_{min} + (N_{max} - N_{min}) \frac{SO_{max} - SO}{SO_{max} - SO_{min}}, \tag{1}$$

$SO$ is the stream size relative measure at the current cell; $SO_{max}$ (whose value depends on the network depth) the same measure at the river mouth and $SO_{min} = 1$ the measure at source cells (namely, cells without any upstream cells, according to the river network). The Manning coefficient is then set to be constant in time while its spatial values decrease towards the river outlet (following the river network), with values between $N_{min} = 0.04$ and $N_{max} = 0.06$.

All these parameters are eventually essential to estimate the spatially- and time-varying average cross-sectional flow velocity in the surface reservoir $v(t)$ following the Manning formula (Manning, 1891):

$$v(t) = \frac{s^{\frac{1}{2}}}{N} \left( \frac{W h_S(t)}{W + 2h_S(t)} \right)^{\frac{2}{3}},$$
(2)

where $h_S$ is the river water depth estimated from the river storage $S$ by

$$h_S = \frac{S}{\rho W L},$$
(3)

and $\rho$ the water density. The flow velocity is ultimately used to estimate the discharge leaving the CTRIP cell:

$$Q_{\text{out}}^S(t) = \frac{v(t)}{L} S(t).$$
(4)

As the definition of most of these parameters is based on empirical relationships, we have to be aware that they inevitably have substantial uncertainties.

## 2.3 CTRIP implementation over the Amazon basin

In this study, we present an OSSE test case over the Amazon river basin whose hydrology is carefully described in Molinier et al. (1993); Wisser et al. (2010). This choice was motivated as the present work follows and complements studies over the same domain (Emery et al., 2016, 2018).

For ISBA-CTRIP, the Amazon basin is composed of a total number of 2028 cells. Based on the basin geomorphology and hydrology (Meade et al., 1991), the basin has been split into 9 spatial regions. These zones, illustrated in Figure 1b, were initially introduced in (Emery et al., 2016) and will be re-exploited here within the application of data assimilation. For a detailed description of the zones, the reader can refer to (Emery et al., 2016).

## 2.4 ISBA-CTRIP forcing

For the present study, ISBA-CTRIP needs external atmospheric forcing in order to run. Similarly to Emery et al. (2016), such data is provided by the Global Soil Wetness Projet 3 (GSWP3, http://hydro.iis.u-tokyo.ac.jp/GSWP3) at a 3-hourly time resolution.

## 3 Method: Synthetic parameter estimation on ISBA-CTRIP

### 3.1 OSSE framework

In OSSE, we introduce beforehand a reference configuration for the model input parameters that we will consider thereafter as the *truth*. From those true parameters, we directly deduce the *true run* from a ISBA-CTRIP model integration. The synthetic observations used for data assimilation are obtained from perturbing the true observables (variables that are used as observations) using an error model that is representative of the real observation errors. The control variables (the model variables to be

corrected with data assimilation) first guess are obtained by directly perturbing the true control variables. Control error also has to be chosen to be representative of the real modeling errors. OSSEs are prerequisite tests to ensure that the implementation of the EnKF algorithm is correct and adapted to the hydrologic problem under consideration (temporal/spatial length-scales, sources of uncertainty, observation operator...).

## 3.2 Data assimilation variables

### 3.2.1 Observation variables and their errors

The observation vector, denoted $\mathbf{y}_k^o$ at the assimilation cycle $k$, is composed of the $n_y$ available observations at cycle $k$:

$$\mathbf{y}_k^o = \begin{bmatrix} y_{k,1}^o, & y_{k,2}^o, & \ldots, & y_{k,n_y}^o \end{bmatrix} \tag{5}$$

where $y_{k,j}^o$, $j = 1 \ldots n_y$, is the $j$-th observation among the $n_y$ at cycle $k$.

In the present study, the observed variables are water depths issued from a simplified SWOT simulator. Note that this simulator will produce water depths while the real SWOT satellite will provide water elevations. As in Biancamaria et al. (2011) and Pedinotti et al. (2014), this SWOT simulator replicates SWOT spatio-temporal coverage. At a given date, the simulator selects the ISBA-CTRIP cells contained (at least 50% of their area) in the SWOT ground tracks. Figure 2 shows some selected ISBA-CTRIP cells under the real swaths over the Amazon basin. The true run is used as a basis to get the true water depths $\mathbf{y}_k^t$. Then, in order to generate the observation vector $\mathbf{y}_k^o$ from the extracted true water depths, each of them is randomly perturbed by adding a white noise characterized by a standard deviation $\sigma^o$ so that:

$$\forall\, j = 1 \ldots n_y,\ y_{k,j}^o = y_{k,j}^t + \epsilon_j^o,\ \epsilon_j^o \simeq \mathcal{N}(0, \sigma^o). \tag{6}$$

Using water depth observations is a strong simplification of the real SWOT product. Therefore, in order to take into account that SWOT will provide water elevations and not directly water depths, this study will look at the assimilation of both water depths and water anomalies. The method for generating these anomalies will be further detailed in Section 4.2.

The observation error is the addition of the measurement error and the representativeness error. The first is associated with inherent instrumental errors when processes are observed and the second represents the error introduced when the observed and simulated variables are not exactly the same (in nature or scale). Following the SWOT uncertainty requirements (Esteban Fernandez, 2017), SWOT-like water surface elevation measurements have a vertical accuracy of 10 cm (when averaged over a water area of 1 km$^2$). This uncertainty accounts for measurement errors due to the remotely-sensed acquisition such as instrumental thermal noise, speckle, troposphere and ionosphere effects. Moreover, we omit error correlations along the swath so that observation errors follow a white noise model. Accounting for spatially-correlated observation errors is an active research area in the field of data assimilation (Guillet et al., 2018) which is beyond the scope of demonstrating the feasibility of assimilating SWOT-type data. In the framework of OSSE, observed and simulated water depths have the same scale as the ISBA-CTRIP model which is used to generate both. In the following, we assume therefore that there is no representativeness error related to the scale in the system. However, it is worth acknowledging that we should expect higher errors on water depths, compared to

water elevations, as we do not know the bathymetry. Assimilation of water depths is performed as a benchmark, against which assimilation of water anomalies will be compared. Ultimately, $\sigma^o$ is chosen as being equal to 10 cm for all observed variables (i.e. both water depths and water elevation anomalies).

### 3.2.2 Control, observation space and their errors

The control vector is denoted by $\mathbf{x}_k \in \mathbb{R}^{n_x}$. It includes the $n_x$ uncertain variables to be estimated through the $k$-th data assimilation cycle. The choice of the control variables determines the observation operator $\mathcal{H}_k$:

$$\mathbf{y}_k = \mathcal{H}_k(\mathbf{x}_k), \tag{7}$$

where $\mathbf{y}_k$ are the simulated observables, in other words, $\mathcal{H}_k$ maps the control variables onto the observation space. They are then compared to the measured observations $\mathbf{y}_k^o$ during the data assimilation experiment. This difference is referred to as the innovation vector.

Following the conclusions from Emery et al. (2016), we determined that assimilating water-depth-like observations would be efficient for the correction of the distribution of the river Manning coefficients. These coefficients are spatially-distributed at the grid-cell scale. However, from Pedinotti et al. (2014), equifinality issues were raised through the correcting of the distribution at this scale. They also affected its upstream-to-downstream spatial distribution.We chose to correct it therefore by applying multiplying factors defined at a coarser scale, namely at the scale of the 9 hydro-geomorphological areas defined in Section 2.3 and illustrated in Figure 1b. Within the same area, the Manning coefficient values are all identically modified by being multiplied by the same factor.

The control vector is composed therefore of the $n_x$ multiplying factors $N_{\mathrm{mult},i}$, $i = 1 \ldots n_x$, applied to the correcting of the spatial distribution of the river Manning coefficient:

$$\mathbf{x}_k = \left[ N_{\mathrm{mult},1} \ldots N_{\mathrm{mult},n_x} \right]^T, \tag{8}$$

giving $n_x = 9$.

The observation operator $\mathcal{H}_k$ maps the control variables (Manning coefficients dimensionless multiplying factors) into the observables (river water depths in meters) as follows:

1. first, apply the multiplying factors $(\mathbf{x}_k)$ to the Manning coefficients distribution;

2. then, apply the ISBA-CTRIP model $\mathcal{M}_{[k-1,k]}$ over the assimilation window $[k-1,k]$ to determine the model states that correspond to the perturbed Manning coefficients distribution;

3. after, turn the CTRIP surface water storage into equivalent water depths following Eq. 3 (we denote by $\mathcal{Z}_k$ the diagnostic operator turning the surface storage variable into the water depth variable);

4. finally, select the simulated water depths under the SWOT swath mask (we denote by $\mathcal{S}_k$ this operator).

The observation operator is therefore the composition of three operators:

$$\mathbf{y}_k = \left(\mathcal{S}_k \circ \mathcal{Z}_k \circ \mathcal{M}_{[k-1,k]}\right)(\mathbf{x}_k) = \mathcal{H}_k(\mathbf{x}_k). \tag{9}$$

Such a non-linear observation operator $\mathcal{H}_k$ is difficult to formulate explicitly which is why we use an EnKF algorithm to estimate the Kalman gain in a statistical way.

## 5   3.3   The EnKF general formulation

In the EnKF framework, the model $\mathcal{M}_{[k-1,k]}$ and observation $\mathcal{H}_k$ operators are generally not linear. The main assumption for the EnKF is to use stochastic ensembles to represent first- and second-order moments (namely the means and the covariances) of the control variable errors (Evensen, 1994, 2003). Indeed, it is assumed that the distribution of the ensemble is similar to that of the error of the control vector and it is also assumed that the Probability Density Function (PDF) of the error is gaussian, thus

well described by its first and second moments. The background control variables $\mathbf{x}_k^b$ (the first guess) are therefore represented by an ensemble of $n_e$ members:

$$\mathbf{X}_{e,k}^b = \begin{bmatrix} \mathbf{x}_k^{b,[1]} & \mathbf{x}_k^{b,[2]} & \dots & \mathbf{x}_k^{b,[n_e]} \end{bmatrix}, \tag{10}$$

To avoid the collapsing of the ensemble, the observation vector in Eq. 5 is randomized by adding a supplementary white noise with the same observation error standard deviation $\sigma^o$ (Burgers et al., 1998) so that

$\forall\, j = 1 \dots n_y,\ \forall\, l = 1 \dots n_e,\ y_{k,j}^{o,[l]} = y_{k,j}^o + \epsilon_j^{o,[l]},\ \epsilon_j^o \simeq \mathcal{N}(0, \sigma^o). \tag{11}$

An observation ensemble is generated:

$$\mathbf{Y}_{e,k}^o = \begin{bmatrix} \mathbf{y}_k^{o,[1]} & \mathbf{y}_k^{o,[2]} & \dots & \mathbf{y}_k^{o,[n_e]} \end{bmatrix}. \tag{12}$$

Note that alternatives exist to the observation randomization chosen here. However, for the present study, we choose to use a full stochastic filter.

Finally, the EnKF analysis step is applied to each member of the ensemble so that

$$\forall\, l = 1 \dots n_e,\ \mathbf{x}_k^{a,[l]} = \mathbf{x}_k^{b,[l]} + \mathbf{K}_{e,k}\left(\mathbf{y}_k^{o,[l]} - \mathcal{H}_k(\mathbf{x}_k^{b,[l]})\right), \tag{13}$$

where $\mathbf{K}_{e,k}$ is the the Kalman gain. It is built from the control and observation error covariance matrices $\mathbf{P}$ and $\mathbf{R}$ and the linearized observation operator $\mathbf{H}$ so that (see Appendix A for more details):

$$\mathbf{K}_{e,k} = [\mathbf{P}\mathbf{H}^T]_{e,k}\left([\mathbf{H}\mathbf{P}\mathbf{H}^T]_{e,k} + \mathbf{R}_k\right)^{-1}. \tag{14}$$

Figure 3 summarizes the general OSSE framework used for the present study. The figure reads from top to bottom and from left to right. An assimilation cycle $[k-1,\ k]$ includes a forecast step where an ensemble of ISBA-CTRIP simulations is integrated, each member having a different spatially-distributed Manning coefficient; an analysis step where the ensemble of Manning coefficients is corrected using synthetic observations through the Kalman filter update in Eq. 13; and a cycling step where the ISBA-CTRIP model is re-run with these analysis estimates to obtain updated model states.

### 3.4 SWOT-based data assimilation special feature

#### 3.4.1 Choice of the assimilation window

We use a 21-day assimilation window corresponding to a SWOT orbit revisit period. During one assimilation window, every pixel under the observation mask is therefore observed at least once. However, its implies that new observations are available at times which differ from the update time. Such a case has already been addressed in several studies. The Ensemble Kalman Smoother (EnKS) for example, introduced by Evensen and Leeuwen (2000), is a direct extension of the EnKF. It consists ofgenerating an update of the control variables taking into consideration the present and all past observations when a new observation is available. The EnKS is actually a sequential version of the Ensemble Smoother (Leeuwen and Evensen, 1996). The latter takes into consideration all past and future observations but turned out to be less effective than the EnKF and the EnKS (Evensen, 1997; Evensen and Leeuwen, 2000). Alternatively, Hunt et al. (2004) developed the 4D-EnKF (4D as in the 4D-VAR variational assimilation methods; Talagrand and Courtier, 1987), which also assimilates observations available at different time-steps. In the 4D-EnKF, all model observations are expressed as a linear combination of the model observations at analysis time and the problem is transformed into a classic EnKF problem. Similarly, Hunt et al. (2007) also presented an asynchronous version of the Local Ensemble Transform Kalman Filter (Bishop et al., 2001; Ott et al., 2004). In the framework of the present study, we apply an Asynchronous Ensemble Kalman Filter (AEnKF) as described by Sakov et al. (2010) and Rakovec et al. (2015). The principle is to increase the dimension of the state in order to consider observations at past and analysis times. This increases the dimension of the matrices which contain covariances between observations available at different times. To our knowledge, the AEnKF has not been used for parameter estimation, as Sakov et al. (2010) and Rakovec et al. (2015) described the method for state estimation experiments.

#### 3.4.2 The Asynchronous EnKF

To start with, $k$ represents the assimilation cycle index but it needs to be distinguished from the day index (within the assimilation cycle) which is the time unit for the observations. We will then denote by $k(i)$, $i = 1 \ldots 21$, the $i$-th day in the current assimilation cycle.

On the $i$-th day of the $k$-th assimilation cycle, the $n_{y,k(i)}$ observations are gathered in the vector $\mathbf{y}^o_{k(i)}$. The overall observation vector at cycle $k$, $\mathbf{y}^o_k$, then concatenates the 21 daily observation vectors $\mathbf{y}^o_{k(i)}$ so that:

$$\mathbf{y}^o_k = \left[ \mathbf{y}^{o,T}_{k(1)} \; \cdots \; \mathbf{y}^{o,T}_{k(21)} \right]^T \in \mathbb{R}^{n_{y,k}}, \; n_y = \sum_{i=1}^{21} n_{y,k(i)}. \tag{15}$$

Similarly to the observation vector, the overall observation operator at the $k$-th cycle, $\mathcal{H}_k$, is the concatenation of the daily observation operators $\mathcal{H}_{k(i)}$, defined from Eq. 9, but by considering the operator $\mathcal{M}$ integrating the model between $k(0)$ ($= (k-1)(21)$) and $k(i)$ as well as the diagnostic and selection operators a time step $k(i)$, $\mathcal{Z}_{k(i)}$ and $\mathcal{S}_{k(i)}$.

The observation error covariance matrix $\mathbf{R}_k$ is the concatenation of the daily observation error covariance matrices:

$$\mathbf{R}_k = \begin{pmatrix} \mathbf{R}_{k(1)} & & & \\ & \mathbf{R}_{k(2)} & 0 & \\ & 0 & \ddots & \\ & & & \mathbf{R}_{k(21)} \end{pmatrix} \text{ with } \mathbf{R}_{k(i)} = (\sigma^o)^2 \mathbf{I}_{n_{y,k(i)}}, \tag{16}$$

where $\mathbf{I}_{n_{y,k(i)}}$ is the identity matrix of size $n_{y,k(i)} \times n_{y,k(i)}$. It turns out that:

$$\mathbf{R}_k = (\sigma^o)^2 \mathbf{I}_{n_y}. \tag{17}$$

Following the same equations as the EnKF, the AEnKF generates, for each member $l = 1 \ldots n_e$ of the control ensemble, an analysis control vector $\mathbf{x}_k^{a,[l]}$.

### 3.4.3    Generation of the ensemble

To generate the background control ensemble, we solely stochastically perturb the variables within the control vector. Note that it amounts to the assumption that all other features of the forward model, e.g. the atmospheric forcings, the LSM structure

and therefore the surface and sub-surface runoff, are perfect. While this applies for OSSEs, such features are never perfect in real-case experiments. This assumption is further discussed in Section 7.

The ensemble of background control vectors $\mathbf{X}_{e,k}^b$, of size $n_x \times n_e$, is generated so that $\mathbf{x}_k^{b,[l]}$, $l = 1 \ldots n_e$, follows a gaussian law of mean $\overline{\mathbf{x}_k^b}$ and covariance matrix $\mathbf{P}_{e,k}^b$. For the first assimilation cycle, the control variables mean value $\overline{\mathbf{x}_1^b}$ is arbitrarily chosen as the openloop run input parameter (the openloop or free run is the model run without assimilation) and the background

error covariance matrix $\mathbf{P}_{e,1}^b$ is a diagonal matrix defined as

$$\mathbf{P}_{e,1}^b = \left( (\sigma^b)^2 \mathbb{I}_{n_x} \right)$$

with $\mathbb{I}_{n_x}$ the identity matrix of size $n_x \times n_x$ and $\sigma^b$ the vector that gathers the initial control variable error standard deviation.

Once the analysis ensemble $\mathbf{X}_{e,k}^a$ is determined, the next step is to propagate the correction in time. In a PE framework, it is necessary to re-run the ensemble model runs during the current assimilation window with the analysis parameters as inputs.

Then, the contribution of the updated parameters is propagated through the model, up to the end of the current assimilation window and put into the model initial condition for the next assimilation cycle.

For the next assimilation cycles, the background mean estimate is set equal to the analysis mean estimate from the previous cycle:

$$\overline{\mathbf{x}_k^b} = \overline{\mathbf{x}_{k-1}^a}.$$

There are different ways of defining $\mathbf{P}_{e,k}^b$. One could choose to stochastically estimate $\mathbf{P}_{e,k-1}^a$ from the analysis ensemble at the previous cycle and use it as $\mathbf{P}_{e,k}^b$. Contrary to state estimation experiments where the analysis error covariance matrix is propagated in time using the model along with the control variables, parameter estimation experiments use it directly as the

background error covariance matrix as there is no dynamical model for the Manning coefficient. The issue with this approach is that the analysis ensemble variance can be strongly reduced and provide too small an ensemble spread to have efficient AEnKF updates in time. To ensure that enough uncertainty is maintained in the ensemble, one can maintain the initial background error covariance matrix through all cycles or impose a minimal value for the variance elements (see Section 5.4).

5    The background error cross-covariance matrix $[\mathbf{PH}^T]_{e,k}$ and covariances matrix $[\mathbf{HPH}^T]_{e,k}$ are directly built from the definition suggested by Evensen (2004), Moradkhani et al. (2005) and Durand et al. (2008), see Appendix A for more details. The matrices are of size $n_x \times n_{y,k}$ and $n_{y,k} \times n_{y,k}$ respectively. The elements in the error cross-covariance matrices result directly from the characterization of the background ensemble, namely the parameter uncertainties accounted for for the generating of the control matrix $\mathbf{X}_{e,k}^b$ and $\mathcal{H}(\mathbf{X}_{e,k}^b)$.

## 10    4    Assimilation strategy

In the incoming experiments, the true control variables $\mathbf{x}^t$ are:

$$\mathbf{x}^t = \begin{bmatrix} 1.65 & 0.85 & 0.85 & 0.95 & 0.90 & 0.95 & 0.90 & 1.30 & 1.40 \end{bmatrix} \tag{18}$$

and the a priori values at the first assimilation cycle $\overline{\mathbf{x}_1^b}$ are:

$$\overline{\mathbf{x}_1^b} = \begin{bmatrix} 1.50 & 0.50 & 0.50 & 0.50 & 0.50 & 0.50 & 0.50 & 1.50 & 1.50 \end{bmatrix}. \tag{19}$$

15    We increase the Manning value in mountainous zones (zones 1 in the Andes and zones 8 and 9 over the shields) and lower the Manning value over the other zones with the lowest values (in zones 2 and 3) corresponding to the main stem. Both true and background values were chosen accordingly.

### 4.1    Sensitivity tests

During one EnKF assimilation cycle, the analysis potentially depends on the following parameters: model spin up, time period
20    (high/low flow), size of the ensemble, control error. Note that the observation error also has an impact on the analysis but its value is already fixed for all subsequent experiments (see Section 5).

A first set of experiments (either model runs or data assimilation runs) will serve as sensitivity tests for the data assimilation platform with respect to the above features. During these sensitivity tests, the different features are tested individually. Table 1 details the range of variations for each tested feature.

### 25    4.2    Assimilation tests

Following the sensitivity tests, a set of three data assimilation experiments will be run and is presented in Table 2. The data assimilation experiments are divided into two categories: the first one uses water depths as observations,the second considers water depth anomalies. All experiments are run across a year, corresponding to 17 assimilation cycles of 21 days.

The first experiment, denoted as PE1, is configured from the aforementioned sensitivity test outcome. The parameters defining the experiment (spinup, starting date, ensemble size, control error) will be those which provide the best results in the sensitivity tests in Table 2. The reference level between the observed and simulated water depths is also the same. In other words, there is no bias in the observation. This first idealized experiment serves as proof-of-concept as the observations type matches exactly the type of the simulated variables. Consequently, with this experiment, we expect to retrieve the true value of the control variables and hence the correct water depths and discharges.

The next step is to head towards more realistic experiments by including new sources of uncertainties in the data assimilation system and seeing how to address them. In this context, two additional experiments denoted as PE2 and PE3 will be carried out. As an example of new uncertainties, SWOT will in fact observe water elevations (water surface elevation as referenced to a geoid or an ellipsoid) whilst CTRIP produces water depths (water surface elevation as referenced to the bottom of the river bed). To perform data assimilation, one needs to convert CTRIP water depths ($h_S^{\text{CTRIP}}$) into CTRIP water elevations ($H_{\text{alti}}^{\text{CTRIP}}$) or inversely for SWOT. It is highly plausible that this operation induces a bias between the modeled and observed water elevations. A simplified example of such situation is illustrated in Figure 4. In this case, SWOT catches the right water elevation dynamic (as $h_S^{\text{SWOT}}$ and $h_S^{\text{CTRIP}}$ are equal) but the direct assimilation of SWOT water elevation $H_{\text{alti}}^{\text{SWOT}}$ will induce a bias as the elevations of the river bed ($H_{\text{bed}}^{\text{SWOT}}$ and $H_{\text{bed}}^{\text{CTRIP}}$) are different between CTRIP and SWOT.

A solution for the handling of this issue is to assimilate water depth anomalies instead of water depths. The next data assimilation experiments, denoted as PE2 and PE3, will therefore test the feasibility of assimilating anomalies. In these experiments, the water depth anomalies are generated by subtracting a time-averaged reference water depth from the current water depth. For all runs (true, openloop or analysis), this time-averaged reference water depth is computed as the mean (true, openloop or analysis) water depth over the year before the start of the assimilation window. It is therefore different for each member of the ensemble. Firstly, in experiment PE2, there will still be no bias between the observed and simulated river bathymetry to observe how the assimilation of anomalies performs. Similarly to PE1, we expect this experiment to be able to retrieve the true control and state variables. Finally, the last experiment PE3, which introduces a constant relative bias between CTRIP and SWOT, will be carried out. For this experiment, we anticipate that the assimilation will still be able to retrieve the model states variables. The use of anomalies as observations should limit the impact of the inserted bias. We do not exclude however that it may be slightly echoed on the control variables.

## 5 Assimilation sensitivity tests

### 5.1 Model spinup sensitivity tests

The objective of the spinup sensitivity tests is to evaluate the minimum spinup period required by the model before applying data assimilation. For this purpose, the model is run several times across two years, from December 19th, 2006 to December 22nd, 2008, corresponding to 35 windows of 21 days (735 days).

A first simulation is run using the true Manning spatial distribution (see Eq. 18) over the two-year time period. We then run 18 additional simulations over the same period with a varying length of the spinup period (see Table 1). Initially, the

simulation setup corresponds to the openloop configuration with the openloop Manning spatial distribution (see Eq. 19). At a given time during the first year, the Manning spatial distribution is instantaneously changed to the true distribution (see Eq. 18) and the model is run until the end of the two years with the true Manning spatial distribution. Table A1 summarizes for each run, the date when the Manning coefficients are changed. The spinup period (expressed as a number of windows of 21 days)

corresponds to the period between when the Manning distribution is changed and the start of the second year, i.e. January 1st, 2008.

To evaluate the spinup impact, the relative difference between the reference run and the test runs is evaluated over the second year of simulation (from January 1st to December 22nd, 2008) and averaged over every window of 21 days. Figure 5 presents the results for the spinup sensitivity test. Each test run (on the x-axis) is identified by its corresponding spinup period length

(expressed as the number of windows of 21 days, see Table A1). We then count (on the y-axis) the number of 21-day windows during which the relative error between the test run and the reference run is higher than a given threshold. We assume that the spinup period is long enough when this number is equal to 0. This number is evaluated from the basin-averaged relative difference and from the relative difference at the downstream station of Obidos, both in terms of water depth and discharge. Note that in Figure 5, when the number of spinup windows of 21 days is equal to 4 on the x-axis, the change from the openloop

Manning distribution to the true one is imposed on October 9th, 2007. Similarly, when this number is equal to 10, the change is imposed on June 5th, 2007. Note also that two thresholds are considered, 0.01 and 0.001. Basin-averaged results are not sensitive to this threshold. There are some differences at Obidos; however, we retain the basin-averaged results to evaluate the required model spinup period.

From all of the results we conclude that a minimum spinup period of 4 windows of 21 cycles, i.e. 84 days, is required. This

period corresponds to the basin concentration time or, in other terms, the required time for the river network to totally empty. In the following sensitivity tests, the model runs start on October 9th, 2007, to be consistent with these results.

## 5.2 Data assimilation starting date sensitivity tests

To evaluate the impact of the starting date, a set of 17 one-cycle-long data assimilation experiments is carried out over the second running year. All experiments have the same general configuration except for the initial date starting from January 1st,

2008 and shifted by 21 days until December 22nd, 2008. This means that the last experiment starts on December 2nd, 2008. The performance of each experiment is evaluated by simply evaluating the spatial average difference between the analysis and the true Manning coefficients. Results presented in Figure 6a-b indicate that there are no significant differences between the data assimilation experiments (for all 17 experiments, the error of the updated Manning coefficients with respect to the true value of the coefficients is below 5 %).

Note that the results in Figure 6a-b show a slight increase in the errors for the experiment starting at the end of the year, presently from September 9th to December 2nd, 2008. This period corresponds to the low-flow season in the Amazon hydrological cycle. Concerning the sensitivity analysis results (Emery et al., 2016), the water depths showed a very low sensitivity to the Manning coefficient during the low flow season. As a consequence of data assimilation, the EnKF is less effectivein low

flow seasonsin the correcting of the Manning coefficient. The analysis relative error (in Figure 6a) and the analysis ensemble dispersion (in Figure 6b) is therefore higher in the low flow season.

For all of the following sensitivity tests, we are only considering therefore one-assimilation-cycle experiments, which will start on January 1st, 2008.

## 5.3 Ensemble size sensitivity tests

The next sensitivity test is dedicated to the ensemble size $n_e$, a critical parameter of any EnKF algorithm. This parameter has to be high enough so as to accurately estimate the Kalman gain matrix but low enough to limit the computational cost (the higher $n_e$, the more model runs are required over each data assimilation window to obtain the analysis estimate of the Manning coefficients).

We consider different ensemble sizes through a one-assimilation-cycle experiment, $n_e$ varies between 10 and 200. Figure 6c compares the analysis Manning coefficient relative error for each ensemble size $n_e$ as in Figure 6a. Results show that the analysis relative error decreases when the ensemble size $n_e$ increases. For an ensemble size $n_e$ equal to 20, the analysis error is below 5 %. Also, for an ensemble size $n_e$ higher than 50, the analysis relative error has converged to a constant value while the analysis ensemble dispersion showed in Figure 6d stabilises.

These results indicate that the ensemble size for future data assimilation experiments should be at least equal to 20; we consider $n_e = 25$ in the present study to limit computational time.

## 5.4 Model error standard deviation sensitivity tests

In this study,we only consider parameter estimation, implying that the background error covariance matrix is associated with the parameter space. We assume that the errors in the Manning coefficients are independent so that the background error covariance matrix is initially specified as a diagonal matrix, where all diagonal elements correspond to the error variances in the spatially- varying Manning coefficients and are equal to the same variance $(\sigma^b)^2$, where $\sigma^b$ is the background error standard deviation. Similarly to previous sensitivity tests, we study here the sensitivity of the data assimilation results to the value of $\sigma^b$ through a one-assimilation-cycle experiment. Figure 6c shows, in logarithmic-scale on the x-axis, the relative error of the updated Manning coefficient with respect to the true coefficient. The analysis error curve shows a decreasing behavior until $\sigma^b$ is in the order of 0.4. It then increases again.

Note that the actual Manning coefficient error before data assimilation is equal to 0.33 (see the blue curve in Figure 6c showing the openloop Manning coefficient error). Consistently, the best data assimilation results are obtained when $\sigma^b$ provides a good approximation of the real error standard deviation. Note also that when $\sigma^b$ becomes too small, data assimilation is less effective. The EnKF algorithm is known to be under-dispersive. Therefore, for future data assimilation experiments, when updating the error covariance matrix from one cycle to another, we will need to make sure that the ensemble dispersion is high enough to cover possible model behavior over the forecast time window by imposing a minimum value for the error variance. Given the sensitivity test results, the minimum value for $\sigma^b$ is set to 0.005. Thus, in the following data assimilation experiments,

we use the analysis error covariance matrix as the background error covariance matrix for the next assimilation cycle, while applying the minimum threshold value on the matrix diagonal terms.

## 6 Data assimilation results

We now present the results from the data assimilation experiments described in Table 2 and in Section 4.4.2. Recall that these experiments aim at correcting a set of 9 multiplying factors applied to the Manning coefficients distribution and constant over 9 hydro-geomorphological zones dividing the Amazon basin.

### 6.1 Assimilation of water depths (PE1)

Figure 7 gives, for each zone, the time evolution of the mean analysis control variable (red) with its dispersion (even though it is very narrow) compared to the truth (black) and the first guess (blue). Similarly, Figure 8 shows, for each zone, the time evolution of the analysis water depth (red) compared to the truth (black) and the openloop (blue). To generate one plot per zone, we use, for each time step, the ensemble of water depths over all cells in the zone and estimate the median value, the first decile and the ninth decile. Furthermore, zone-averaged normalized Root Mean Square Error (RMSEn) statistics are given in Tables A2 and A3 in Appendix D.

In general, the PE1 experiment gives very good results as the analysis mean for each zone retrieves the true value with a very low dispersion. However, the data assimilation algorithm features spatially-dependent behavior, see Figure 7):

- Firstly, the control variable for the zones 1, 2, 3, 4, 5 and 9 converges instantaneously (in only one assimilation cycle) toward the true values and remains at these true values for all following cycles.

- A similar behavior can be observed for zones 6, 7 and 8 from the first cycle to around the ninth cycle. For the remaining cycles, we notice an increase in the mean analysis estimate, along with the ensemble dispersion, until around the thirteenth cycle and after, a decrease back to the true value.

These observations can be explained with the global sensitivity analysis results for water depths in Emery et al. (2016) and using Figure 8:

- Firstly, zones 1, 2 and 3 correspond to the river main stem, whilst zones 4, 5 and 9 correspond to the main left-bank tributaries, namely the Caquetá/Japurá river (zone 4) and the Negro river (zone 5). In these zones, as the Manning coefficient is directly corrected in the first assimilation cycle, the analysis water depths (red) overlap the true water depth (black). Furthermore, Figure 8 for these zones shows that the openloop (blue) and true (black) water depths have a very similar variability in time but differ by a constant bias. The global sensitivity analysis results in these zones showed a constant first-order sensitivity in time to the Manning coefficient all year long. This first-order sensitivity means that the contribution of the Manning coefficient to the water depth is linear. Correcting the Manning coefficient in these zones equates therefore to correcting the bias between the openloop and the true water depths.

- Subsequently, zones 6, 7 and 8 correspond to right-bank tributaries, namely the Juruá and Purus rivers (zone 6), the Madeira river (zone 7) and the Tapajós and Xingu rivers (zone 8). These right-bank tributary zones are characterized by a strong seasonal cycle (see Figure 8, zones 6-8). By comparing the corresponding plots in Figures 7 and 8, we notice that the period when the analysis control variable spreads from the truth corresponds to the low flow season in these zones. According to the global sensitivity analysis results, water depths in these zones are less sensitive to the Manning coefficient in low flow conditions. Additionally, there is very little water in the zones during this period and consequently, the background control ensemble is not spread out enough for the EnKF to be efficient. Meanwhile, the EnKF still sees that the observations are higher than the model predictions (as seen with the positive innovations in these zones shown in Figure A2). In order to increase the simulated water depth, the EnKF therefore corrects the Manning coefficient so that its value rises (a higher Manning coefficient means a slower flow velocity and then a higher simulated water depth). Finally, once the low flow season ends, the analysis Manning coefficient converges back to the truth (see the last assimilation cycles).

## 6.2 Assimilation of water anomalies (PE2 and PE3)

The assimilation of anomalies has been tested over two experiments denoted PE2 and PE3 (see Table 2). Note that the observation error standard deviation $\sigma^o$ remains equal to 10 cm as, with these experiments, we only aim at testing the feasibility of assimilating water depth anomalies. In the PE2 experiment, there is no difference of bathymetry between the simulated and observed water anomalies whilst the river bankful depth is different in the PE3 experiment. Figure 9 gives, for each zone, the time evolution of the mean analysis control variable for PE2 (orange) and PE3 (purple) with their dispersion compared to the truth (black) and the first guess (blue). Again, zone-averaged normalized Root Mean Square Error (RMSEn) statistics for these experiments are given in Tables A2 and A3 in Appendix D.

The general configuration of the experiments PE1 and PE2 is the same. The only difference between the two experiments is the nature of the observations: water depths for PE1 and water anomalies for PE2. Like experiment PE1, experiment PE2 (the orange line in Figure 9) gives very good results. All control variables converge toward the true values more or less rapidly. The control variable for the zones 4 to 8 instantaneously (in only one assimilation cycle) converges toward the true value while the convergence is slower for the remaining zones as around five assimilation cycles are needed to retrieve the true value. This slower convergence for these zones can be explained by the fact that the magnitude of the observed water anomalies is generally smaller, compared to the water depths assimilated in PE1. The ratio between the observation error and the observations themselves is also then smaller, resulting in a smaller EnKF gain. The control variable correcting increment is smaller for the anomalies therefore than for the water depth and more cycles are needed to converge.

As for the PE3 experiment results - the purple line in Figure 9 - the assimilation still gives good results but not as good as in previous experiments. The control variables still instantaneously converge toward the truth in the zones 4, 5 and 6. Concerning the other zones, there is no clear convergence towards the true value. Instead, the analysis control variables either get closer but remain distinct from the true value (zones 2 and 3) or temporarily deviate from the truth during the experiment (zones 1, 7, 8 and 9). Still, despite the control variables not clearly converging towards the truth, the simulated water depths using the

analysis control variables, presented in Figure 10, display a very low deviation from the truth, confirming the general good performance of the data assimilation procedure.

Comparing the control variables and water depth time variations, it appears that the control variables are deviating from the truth mainly when water depths are decreasing, in between the high flow and low flow seasons. During this period, the model goes from a state where floods occur to a state where there is no flood, particularly in the zones 2-3 and 7-8 with a clear seasonal cycle. On the other hand, no flood event was spotted in zones 4, 5 and 6 where the best results were obtained. In ISBA-CTRIP, the activation/deactivation of the flooding scheme is triggered by the simulated water depth exceeding/becoming lower than the river bankful depth. Yet, in the experiment PE3, this river bankful depth differs between the model and the observation because we artificially inserted a bias between the simulated and observed water depths. More specifically, the river bankful depth is lower in the model than in the observations. Therefore, the control variables deviating from the true value when the water depth is decreasing are indicative of the simulated water depths presenting floods when there is no flood in the observations. The activation of the flood scheme changes the dynamics of the water depth in the river. As part of the water in the river is spilled into the floodplains, water level variations in the river are slower. The flooded model needs then a stronger variation of the Manning coefficient in order to catch the non-flooded observed water level. Ultimately, the stronger variations of the estimated Manning coefficient allow the retrieval of the true water depths.

## 7    Discussions

The results presented here are preliminary investigations into the assimilation of SWOT water surface elevations product into a large-scale hydrological model. This study focused on the correction of a critical river parameter, here the river Manning coefficient.

For all simulations, the Manning coefficient distribution is set to be constant in time. For each grid cell, one value of the Manning coefficient is used for the entire simulation. However, in reality, it is commonly accepted that this parameter could vary in time, depending on the seasonal cycle or also some extreme hydrological event such as large flooding events, which can even modify the bathymetry itself. The results showed that, for this OSSE, the data assimilation is able to converge quite quickly towards the true value. For example, for the left-bank tributaries zones, namely zones 4 and 5, in every experiment, the associated control variable converges toward the true value in only one assimilation cycle. In a real-case experiment, we could expect to retrieve the temporal variations of the Manning coefficient from one assimilation cycle to another. The good performances of the assimilation platform are mainly related to the fact that, in the ISBA-CTRIP model, the water depth diagnostic variables are sensitive to the Manning coefficient (Emery et al., 2016). Simulated water depths are not then that sensitive to the Manning coefficient (e.g. in the right-bank tributary zones during the low flow season), the data assimilation performances slightly degrade. These results are specific to the ISBA-CTRIP model. To apply the same method to another model and even another region, one needs to first study the sensitivity of the (other) model to the (other) study region.

Secondly, the study investigates the potential of assimilating water surface anomalies instead of direct water surface elevations. The use of water surface anomalies is driven by the need to avoid potential bias between the control and the observed

variables. Indeed, a bias will likely be introduced from a discrepancy between the elevation of the river bed in the model and in the observations with respect to a reference surface such as a geoid or an ellipsoid (see Figure 4). Under the assumption that the water variations are the same between the model and the observations, the use of anomalies as observed variables should prevent this bias from affecting the results.

Another likely bathymetry error corresponds to errors on the river bankful depth, the river width and more generally, representativeness errors due to the use of a simplified bathymetry. This type of error was artificially introduced by perturbing the model bankful depth in PE3. Specifically to ISBA-CTRIP, the river bankful depth controls when the model floods, which has a direct impact on the water depth dynamics. Background anomalies and observed anomalies may therefore present different dynamics where either the observed variables flood while the model variables do not or inversely. The experiment PE3 illus-

trated the effect of this bias on the variations of Manning coefficients. Instead of being maintained at their true value, their value slightly varied around the true values to account for the difference in dynamics between the model and the observations. However, one could expect even more variations in the updated control variables around the true value to increase if more and different errors in the bathymetry exist (which will likely happen with more realistic experiments).

Furthermore, real-case experiments may suffer from another type of bias originating from errors in the atmospheric forcing

and in the surface and sub-surface runoff provided by the LSM (i.e. ISBA). Both control the amount of water entering the river system. A basic idea to attenuate this issue would be to consider their uncertainties when generating the background ensemble. This approach may become limited when the errors in the forcing are very large. it may also lead to unrealistic, even non-physical, updated Manning coefficient values. Besides, when correcting the model's parameters, we only re-distribute the water volume within the basin whilst such types of errors could actually require adding/withdrawing water to/from the system.

The potential solution would then be to include such forcing or LSM variables in the control vector or, to update variables closer to the observations, including CTRIP's state variables such as the water storage. This would change the current framework to a dual state-parameter estimation approach.

Noting this, there may be an additional advantage in assimilating water anomalies instead of the direct water depths. Comparing the Kalman gain between the PE1 experiment (which assimilated direct water depths) and the PE2/PE3 experiments

(whicha ssimilated water anomalies), the gain magnitude for the water anomalies is lower than the water depth gain magnitude. This is to be expected as the Kalman gain is stochastically estimated from an ensemble of model runs and the magnitude of the simulated water anomalies is lower than the simulated water depth magnitude. The consequence of this lower gain is that the correction applied to the control variable is also lower. If the convergence towards the true value takes more than one assimilation cycle, the divergence from it in the presence of bias is also diminished.

Beyond the bias issues, real-data assimilation configuration will raise the question of the unknown true parameter value, if it exists. Firstly, there will be no true estimates of the control variables with which to compare the assimilated simulations. The assimilation will be evaluated against the observed variables directly. Then, with the real data, model structure error will be introduced. To our knowledge, the model structure error is still a challenging error to estimate and most data assimilation studies assume no model structure error. However, when using an ensemble-based model, a possibility for dealing with such

structure error is to enrich the background ensemble by considering more uncertainties from variables that are not necessarily

in the control vector (including errors in the forcing or parameters from both the LSM and the RRM). The capacity of such ensembles to tackle model structure errors can be tested using synthetic observations based on a different hydrological model.

Additionally, the real SWOT data will have a finer resolution than the synthetical SWOT data currently used. Still, the coarser resolution observations are found to provide information to constrain the model and to improve the value of the spatially-varying Manning coefficients. Then, when moving to real-data assimilation experiments, we can consider averaging the fine-scale SWOT product over a coarse grid cell corresponding to an ISBA-CTRIP cell so that the resolution of the observations and the model matches.

Ultimately, heading towards more realistic experiments also implies more realistic representations of the observation errors. But more complex errors should be expected for the real SWOT product. Some correlated errors along the swath should be expected due to the instrument but also to the motion of the satellite and delays due to propagation of the electromagnetic waves in the ionosphere and atmosphere. Nevertheless, as part of the mission science requirements, the sum of all errors should not exceed 10 cm when the measured data is averaged over 1 km$^2$. There should also be additional errors affecting the observations that can be described as "detectability errors" such as "dark water", "layover" and "false positive". "Dark water" pixels will result in missing data and will not be included in the assimilation, "layover" pixels will have a higher vertical error due to surrounding vegetation and topography, but should also be flagged (Biancamaria et al., 2016). Eventually, "false positive" pixels (i.e. pixels classified as water, whereas they correspond to land) will be the most complicated to anticipate. With these additional errors taken into account in the assimilation framework, one could expect a slower convergence of the control variables. Note that these aspects of the measurement errors are related to water surface elevation products.

## 8    Conclusions

This study presents a series of OSSE that assimilates SWOT-like synthetic observations of water depths and anomalies into the large-scale hydrological model ISBA-CTRIP in order to correct the spatially-distributed Manning coefficient. The study is applied over the Amazon river basin. Prior to the actual data assimilation experiments, a series of sensitivity tests was conducted to study the sensitivity of the data assimilation performance to the different features of the EnKF, in particular the size of the ensemble. Then, three full-year data assimilation experiments were run based on the outcomes of the sensitivity tests. For all three experiments, the assimilation was able to track back the true value of the Manning coefficient distribution.

The sensitivity tests successively studied the sensitivity of the data assimilation platform to model spinup period, the experiment starting date through the hydrological year, the size of the ensemble for the EnKF and the initial control variable standard deviation. These tests showed first of all that a spin up of four windows of 21 days is sufficient for the transitional period due to a sudden change in the Manning coefficient distribution in the model. The second sensitivity test then demonstrated that the data assimilation performance is not clearly sensitive to the period of the hydrological year when the experiment is done. The next sensitivity test informed us that an ensemble of 25 members was enough to obtain good EnKF performances. Finally, the last sensitivity test studied the effect of the control variable error standard deviation and the best performances were obtained

for prior standard deviation between 0.05 and 0.75, which corresponds to the order of magnitude of the actual error between the true and openloop control variables.

Using these results, we run three data assimilation experiments over approximately one year (the year 2008). The first experiment (PE1) assimilated direct pseudo-observations of water depths. Results showed the capability of the data assimilation algorithm to converge very quickly toward the true value, generally in only one assimilation cycle. Still, during the low flow season, the assimilation was less effective in the zones with a clear seasonal cycle. This was explained by the fact that during this period, water depths are less sensitive to the Manning coefficient.

The two other experiments (PE2 and PE3) introduced and tested the assimilation of water surface anomalies. The anomalies were obtained by subtracting a yearly-averaged water depth from the current water depth in both the model and the observations. The first water anomalies assimilation experiment (PE2) provided very good results with all the control variables also converging towards their associated true values. However, the convergence was slightly slower than during the assimilation of the water depth (between 1 and 5 assimilation cycles). This is explained by a lower Kalman gain when updating the Manning coefficient.

The last experiment also assimilated water anomalies (PE3). For this particular experiment however a bias was artificially introduced in the river bathymetry. For this experiment, the assimilation was still able to get closer to the true value but, for the some zones like the mainstream zones, there was no convergence as the control variables kept varying around the true value. This phenomenon was explained by the detection of floods in the model but not in the observations. Still, the statistics of the Manning coefficient distribution and the simulated water depths after assimilation remain as improved compared to the openloop simulations. Ultimately, these two experiments demonstrated the feasibility of assimilating water surface anomalies to correct the Manning coefficient.

These experiments offer several perspectives. They mainly consist of approaching more realistic data assimilation experiments which take into account more sources of uncertainties between the model and the observations, such as correlated observation errors or uncertainties in the forcing and the LSM surface and sub-surface runoff. To test the platform's limitations regarding the DEM/bathymetry bias issue, one can use simulated water surface elevations referenced to a geoid instead of water depths from the model or even assimilate water depths from another model where the bathymetry differs. As most applications generally require a good estimate of the river flow and river water volume, another lead of investigation could maintain the SWOT-based OSSE framework but correct the simulated water storage and/or discharge, either as a single state estimation framework or as a dual state-parameter estimation framework (similarly to dual discharge-bathymetry inference methods developed by Oubanas et al., 2018 and Brisset et al., 2018 for some hydraulic models). Moreover, along with observations of water surface elevations, SWOT will also provide two-dimensional maps of river widths and surface slopes. One can also study the possibility of assimilating such products to constrain other parameters such as the bankful depth that controls the model flooding scheme.

*Code and data availability.* The CTRIP code is open source and is available as a part of the surface modelling platform called SURFEX, which can be downloaded at http://www.cnrm-game-meteo.fr/surfex/. SURFEX is updated approximately every 3 to 6 months and the CTRIP version presented in this paper is from SURFEX version 7.3. If more frequent updates are needed, please follow the procedure informing you of how to obtain a SVN or Git account in order to access real-time modifications of the code (see the instructions inthe previous link). The

ISBA-CTRIP model is coupled to the DA codes via the OpenPalm coupler available at http://www.cerfacs.fr/globc/PALM_WEB/. To get the DA routines coupled to ISBA-CTRIP with OpenPalm, please directly contact C. Emery (charlotte.emery@jpl.nasa.gov) or S. Biancamaria (sylvain.biancamaria@legos.obs-mip.fr). To obtain the GSWP3 forcings, please refer to the following url: http://search.diasjp.net/en/dataset/ GSWP3_EXP1_Forcing (https://doi.org/10.20783/DIAS.501).

## Appendix A: Definition of error covariance matrices

The background error cross-covariance matrices $[\mathbf{PH}^T]_{e,k}$ and $[\mathbf{HPH}^T]_{e,k}$ are defined based on Evensen (2004); Moradkhani et al. (2005); Durand et al. (2008) so that:

$$[\mathbf{PH}^T]_{e,k} = (n_e - 1)^{-1} \left( \mathbf{X}_{e,k}^b - \overline{\mathbf{X}_{\bullet,k}^b}.\mathbf{1}_{n_e}^T \right) \left( \mathcal{H}(\mathbf{X}_{e,k}^b) - \overline{\mathcal{H}(\mathbf{X}_{\bullet,k}^b)}.\mathbf{1}_{n_e}^T \right)^T, \tag{A1}$$

and

$$[\mathbf{HPH}^T]_{e,k} = (n_e - 1)^{-1} \left( \mathcal{H}(\mathbf{X}_{e,k}^b) - \overline{\mathcal{H}(\mathbf{X}_{\bullet,k}^b)}.\mathbf{1}_{n_e}^T \right) \left( \mathcal{H}(\mathbf{X}_{e,k}^b) - \overline{\mathcal{H}(\mathbf{X}_{\bullet,k}^b)}.\mathbf{1}_{n_e}^T \right)^T. \tag{A2}$$

In those definitions, $\mathbf{X}_{e,k}^b$ is the control matrix storing the $n_e$ control vectors $\mathbf{x}_k^{b,[l]}$, $l = 1 \ldots n_e$, from the background ensemble such that

$$\mathbf{X}_{e,k}^b = \left[ \mathbf{x}_k^{b,[1]} \ \ldots \ \mathbf{x}_k^{b,[N_e]} \right]$$

.

Next, $\mathcal{H}(\mathbf{X}_{e,k}^b)$ represents the same control matrix but mapped into the observation space:

$$\mathcal{H}(\mathbf{X}_{e,k}^b) = \left[ \mathcal{H}(\mathbf{x}_k^{b,[1]}) \ \ldots \ \mathcal{H}(\mathbf{x}_k^{b,[n_e]}) \right].$$

Also, $\overline{\mathbf{X}_{\bullet,k}^b}$ and $\overline{\mathcal{H}(\mathbf{X}_{\bullet,k}^b)}$ are the corresponding ensemble expectations such that

$$\overline{\mathbf{X}_{\bullet,k}^b} = \frac{1}{n_e} \sum_{l=1}^{n_e} \mathbf{x}_k^{b,[l]} \quad \overline{\mathcal{H}(\mathbf{X}_{\bullet,k}^b)} = \frac{1}{n_e} \sum_{l=1}^{n_e} \mathcal{H}(\mathbf{x}_k^{b,[l]}).$$

These vectors dimension are $n_x$ and $n_{y,k}$ respectively. Finally, $\mathbf{1}_{n_e}$ is a vector of size $n_e$ containing only 1s.

## Appendix B: Spinup sensitivity test additional tables

Table A1 summarizes for each run, the date when the Manning coefficients are changed. The spinup period (expressed as a number of windows of 21 days) corresponds to the period between when the Manning distribution is changed and the start of the second year, i.e. January 1st, 2008.

## Appendix C:  Sensitivity tests results per zones

Figure A1 displays the sensitivity tests results (as in Figure 6) but for each zone separately.

## Appendix D:  Assimilation performances at the zones scales

At each grid-cell of the study domain, we estimated the normalized Root Mean Square Error (RMSEn) before and after assimilation by comparing the openloop and the mean analysis simulations respectively to the true simulation, for both the simulated water depth and discharge:

$$\text{RMSEn}_i = \frac{\sqrt{\frac{1}{N}\sum_{n=1}^{N}(V_{n,i}^* - V_{n,i}^t)^2}}{\overline{V_{.,i}^t}}, \tag{D1}$$

where the state variable $V$ is either the discharge or the water depth, $n$ is the time indice, $i$ is the grid-cell indice, the $t$ superscript represents the "truth" and the $*$ superscript represents either the openloop or the analysis ensemble average.

Table A2-A3 gives these statistics for all experiments averaged over each control zone. Table A2 shows the water depths zone-averaged RMSEn and Table A3 shows the discharge zone-averaged RMSEn.

## Appendix E:  Assimilation results additional figures

Figure A2 displays the evolution along the assimilation cycles of the averaged innovations. The sign of the innovation will drive the direction of the correction brought by the assimilation:

- A positive innovation means that the observations are higher than the model. Physically, the simulated flow is too fast and the water leaves the river reservoir too quickly. This means that the river Manning coefficient needs to be increased to slow the flow,

- A negative innovation means that the observations are lower than the model. Physically, the simulated flow is too slow and the water remains in the river reservoir. This means that the river Manning coefficient needs to be increased to accelerate the flow.

*Author contributions.*   Charlotte M. Emery designed and carried out the experiments under the supervision of Sylvain Biancamaria and Aaron Boone, as part of her PhD project. Sylvain Biancamaria provided the SWOT-based observation mask used to generate the synthetic observations and Aaron Boone provided support for the use of the ISBA-CTRIP hydrologic model. Charlotte M. Emery prepared the manuscript with contribution from all co-authors.

*Competing interests.*   The authors declare that they have no conflict of interest.

*Acknowledgements.* This work was supported by the CNES, through a grant from Terre-Océan-Surfaces Continentales-Atmosphère (TOSCA) committee attributed to the project entitled "Towards an improved understanding of the global hydrological cycle using SWOT measurements". C. M. Emery received doctoral research support from CNES/région Midi-Pyrénées grant. C. M. Emery and C. H. David received support from the Jet Propulsion Laboratory, California Institute of Technology, under a contract with NASA; including grants from the

5   SWOT Science Team and the Terrestrial Hydrology Program.

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

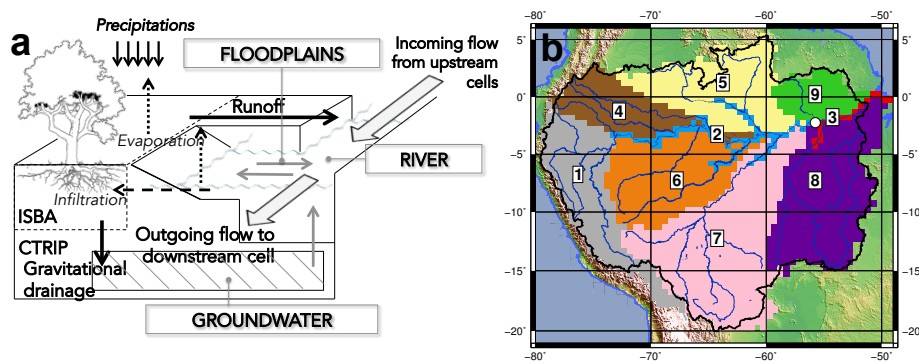

**Figure 1.** (a) The ISBA-CTRIP system for a given grid cell. ISBA surface runoff ($Q_{\text{ISBA,sur}}$) flows into the river/surface reservoir $S$, ISBA gravitational drainage ($Q_{\text{ISBA,sub}}$) feeds groundwater reservoir $G$. The surface water is transferred from one cell to another following the TRIP river routing network. (b) Hydro-geomorphological areas of the Amazon basin from Emery et al. (2016) with the gauge of Óbidos located by the white circle at the entry of the zone 3.

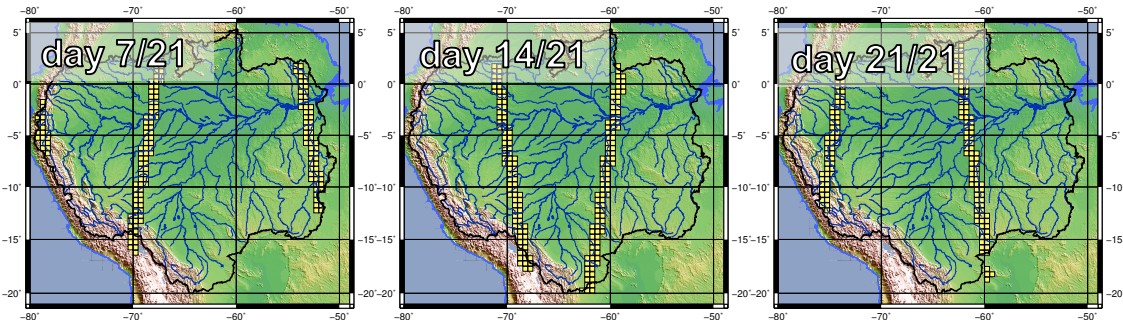

**Figure 2.** SWOT swaths at ISBA-CTRIP resolution over the Amazon basin.

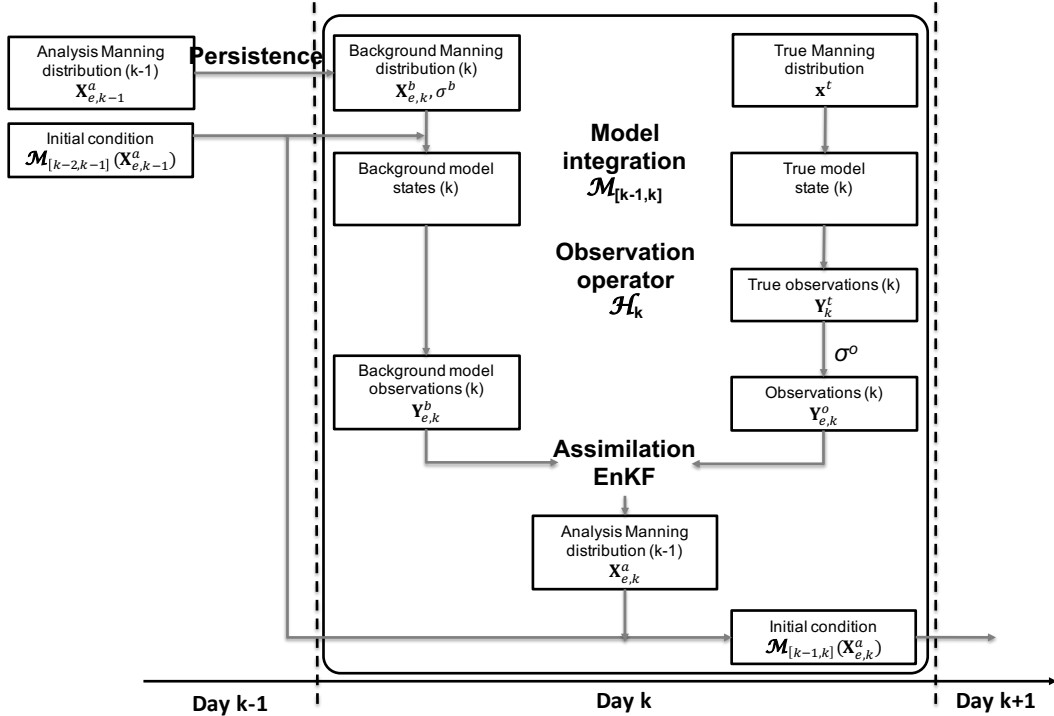

**Figure 3.** Data assimilation framework over the assimilation cycle $[k-1, \ k]$ including (1) a forecast step to integrate the ensemble of ISBA-CTRIP simulations, each member having a different spatially-distributed Manning coefficient, (2) an analysis step to correct this ensemble of Manning coefficients using synthetic observations through the Kalman filter equation and (3) re-run the ISBA-CTRIP model with these analysis estimates to obtain the updated model states.

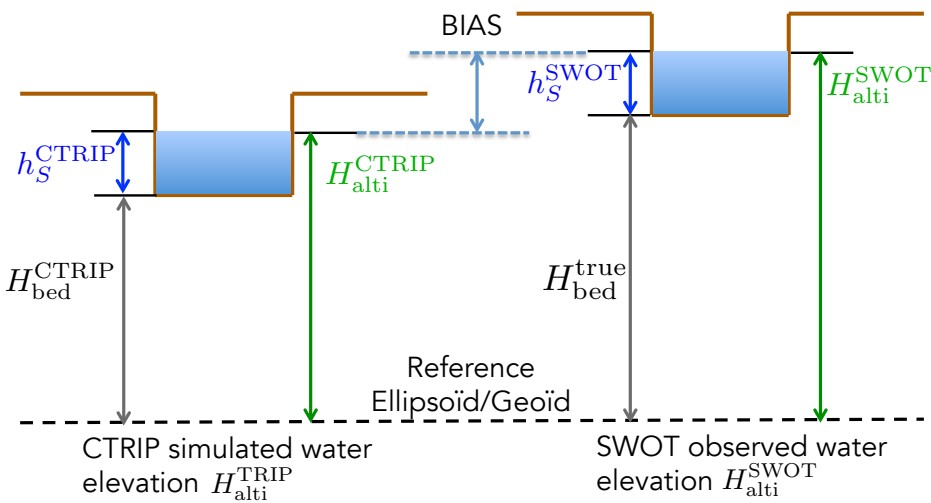

**Figure 4.** Illustration of a bias case between the water elevation as observed by SWOT ($H_{\text{alti}}^{\text{SWOT}}$) and the one simulated by CTRIP ($H_{\text{alti}}^{\text{TRIP}}$) because of a bias between model and true river beds.

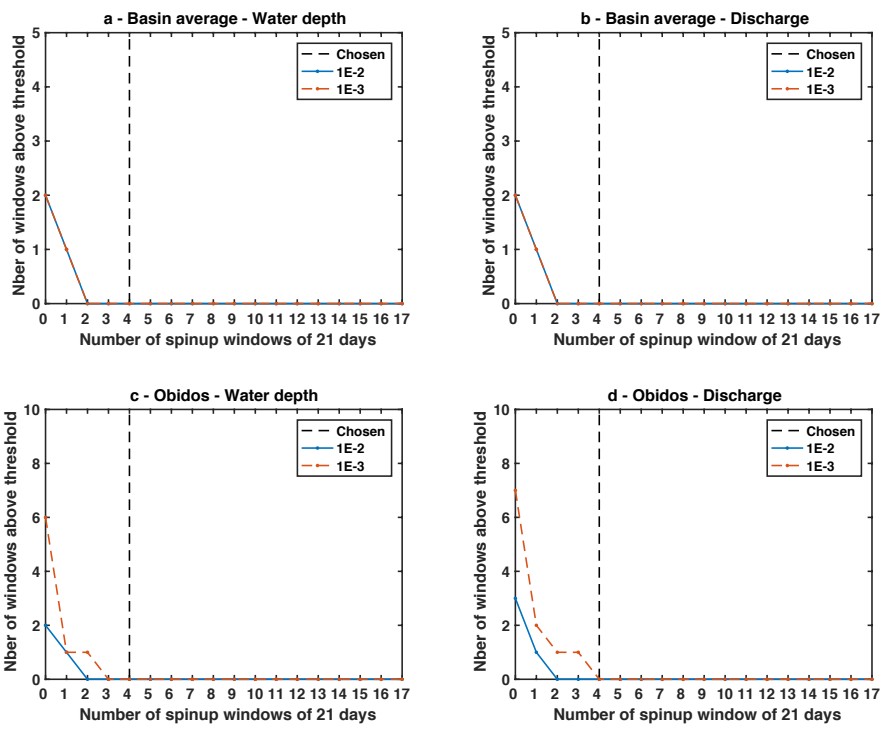

**Figure 5.** Results for the spinup sensitivity test. Each test run is represented along the $x$-axis and referenced by its number of spinup windows. The $y$-axis displays the number of windows during which the relative difference between the true run and the openloop run in which the Manning spatial distribution is modified is above the chosen threshold. These statistics are obtained for the discharge (a,c) and the water depth (b,d) and evaluated over the entire basin (a,b) and at the downstream station of Óbidos (c,d). Note that the vertical dashed line corresponds to the minimum model spinup period retained in this study.

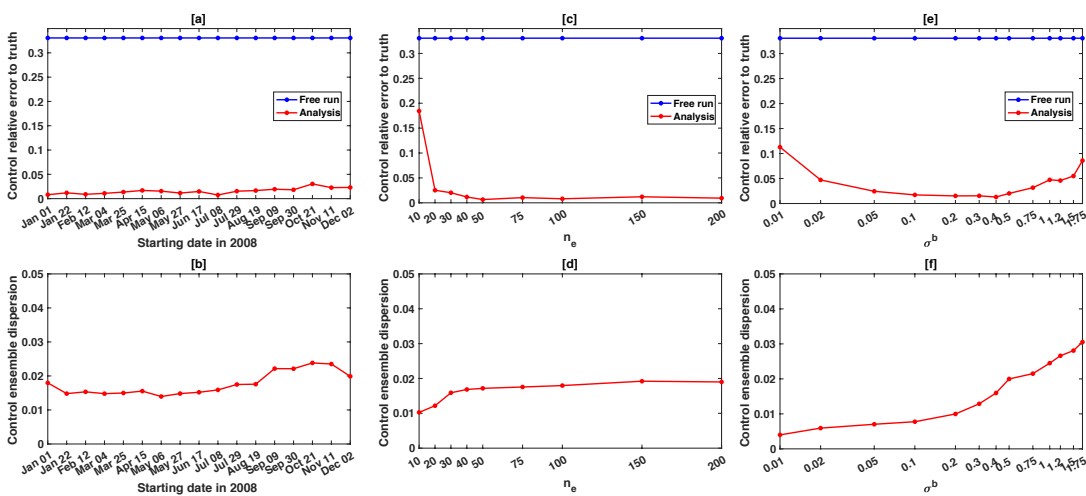

**Figure 6.** (Top) Relative error (to the truth) and (Bottom) dispersion of the analysis control ensemble (averaged over all control variables) for the sensitivity tests to (a-b) the data assimilation starting date, (c-d) the ensemble size $n_e$, (e-f) the background error standard deviation $\sigma^b$. For each test, a set of one-cycle-long data assimilation experiments is run.

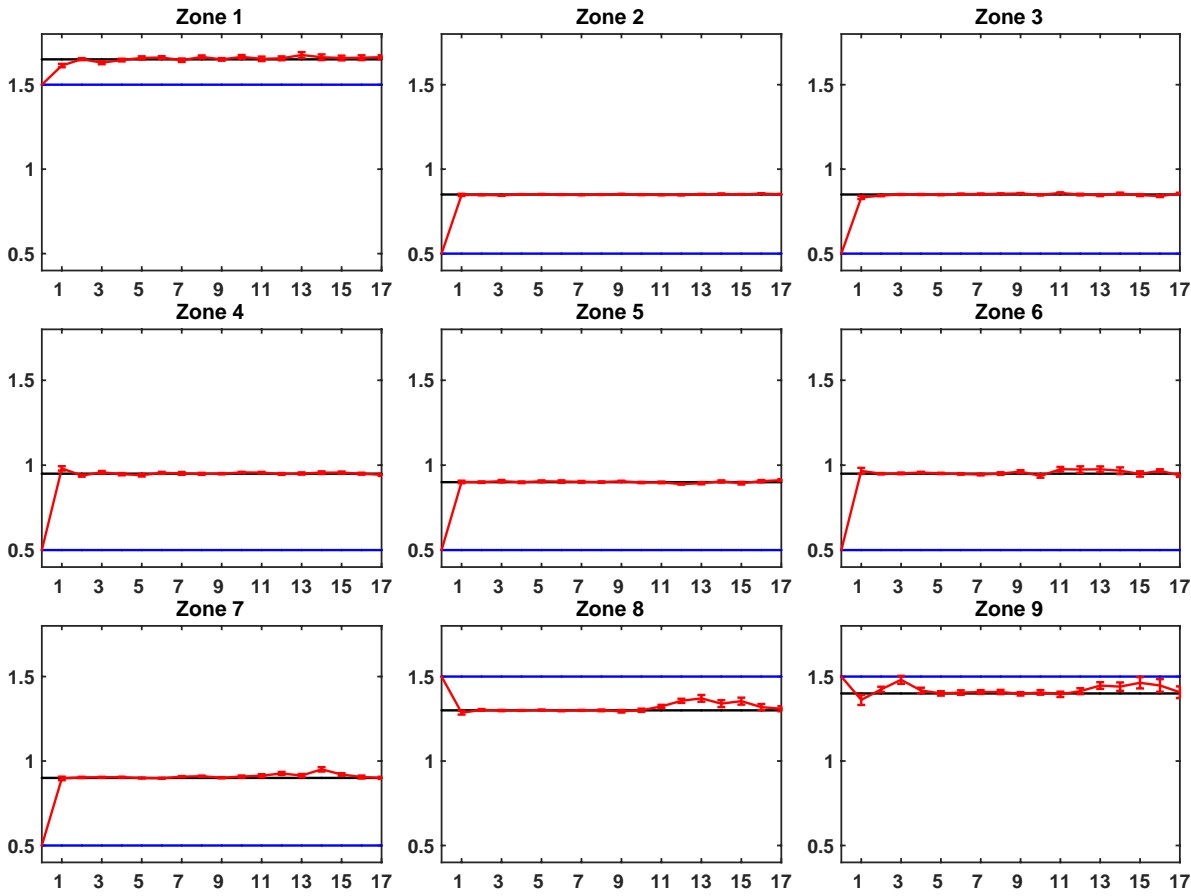

**Figure 7.** Control variables assimilation results for the PE1 experiment: evolution of the ensemble-averaged analysis control variable (red line) for each zone (one zone per subplot) with respect to the assimilation cycle and compared to the corresponding true value (black line) and the openloop value (blue line).

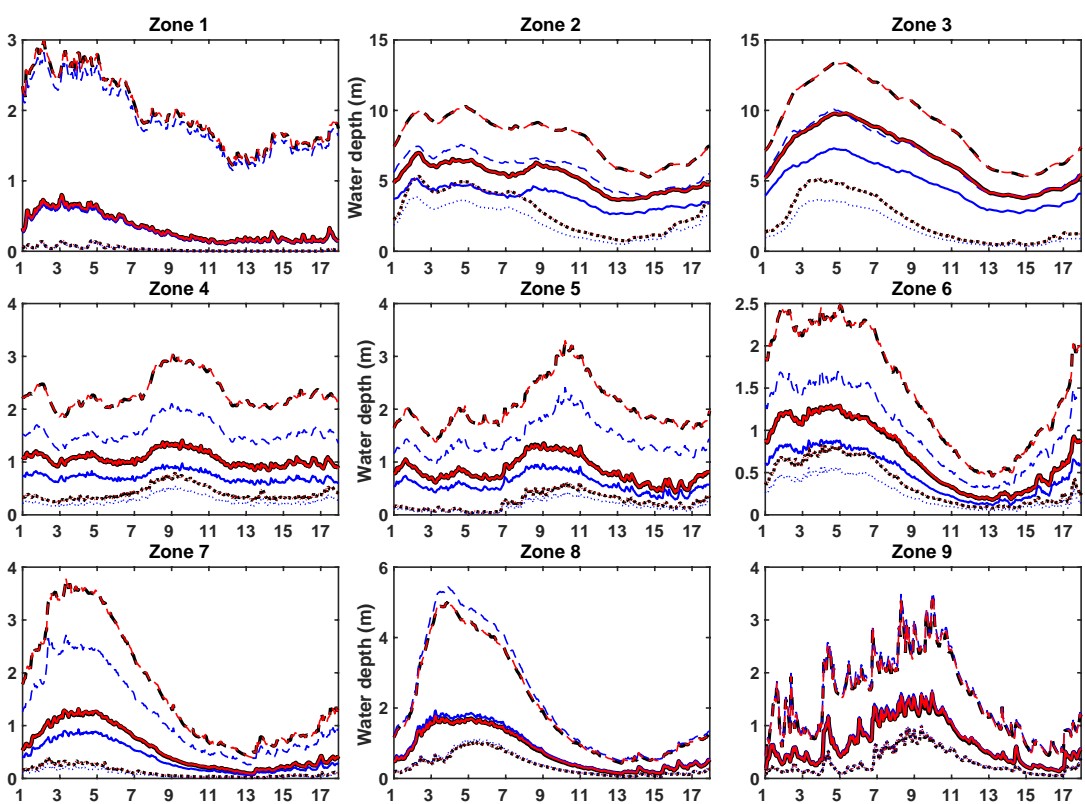

**Figure 8.** Water depths assimilation results for the PE1 experiment: daily evolution of the ensemble-averaged analysis water depth (red lines) compared to the true water depths (black lines) and the openloop water depths (blue line). For each zone (one per subplot), the median (full line), the first decile (dotted line) and the ninth decile (dashed line) of water depth ensemble over all grid cells in the zone are represented.

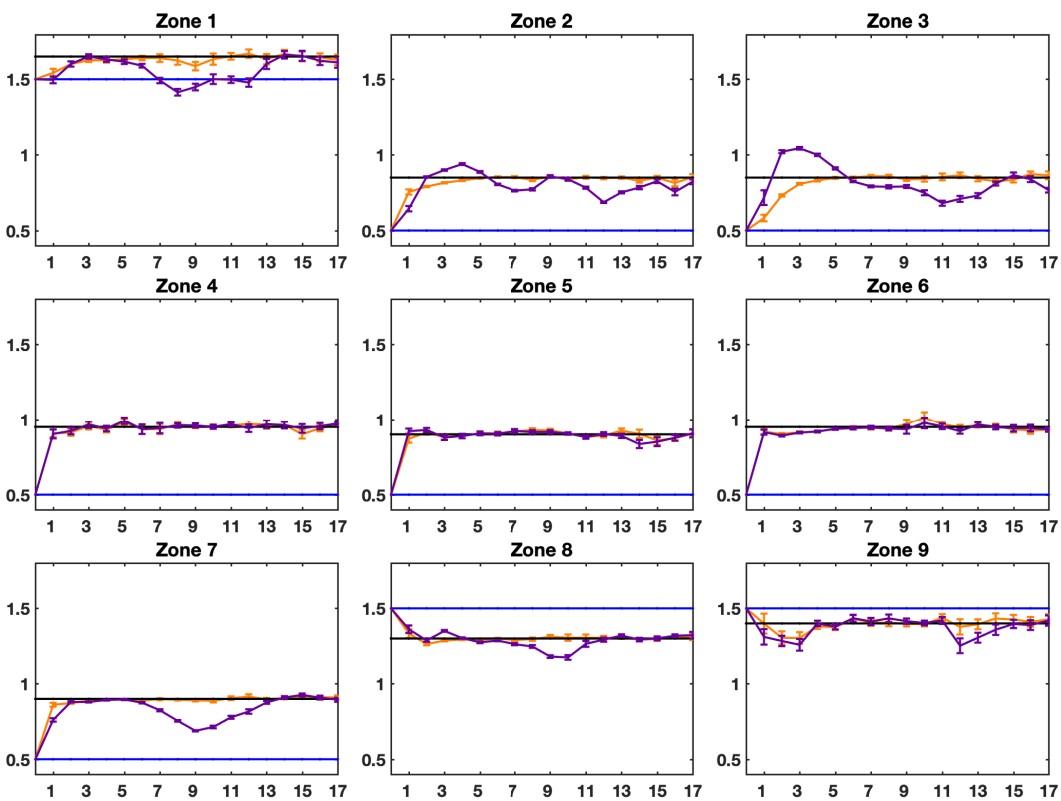

**Figure 9.** Control variables assimilation results for the PE2 and PE3 experiments: evolution of the ensemble-averaged analysis control variable for the PE2 experiment (orange line) and the PE3 experiment (purple line) for each zone (one zone per subplot) with respect to the assimilation cycle and compared to the corresponding true value (black line) and the openloop value (blue line).

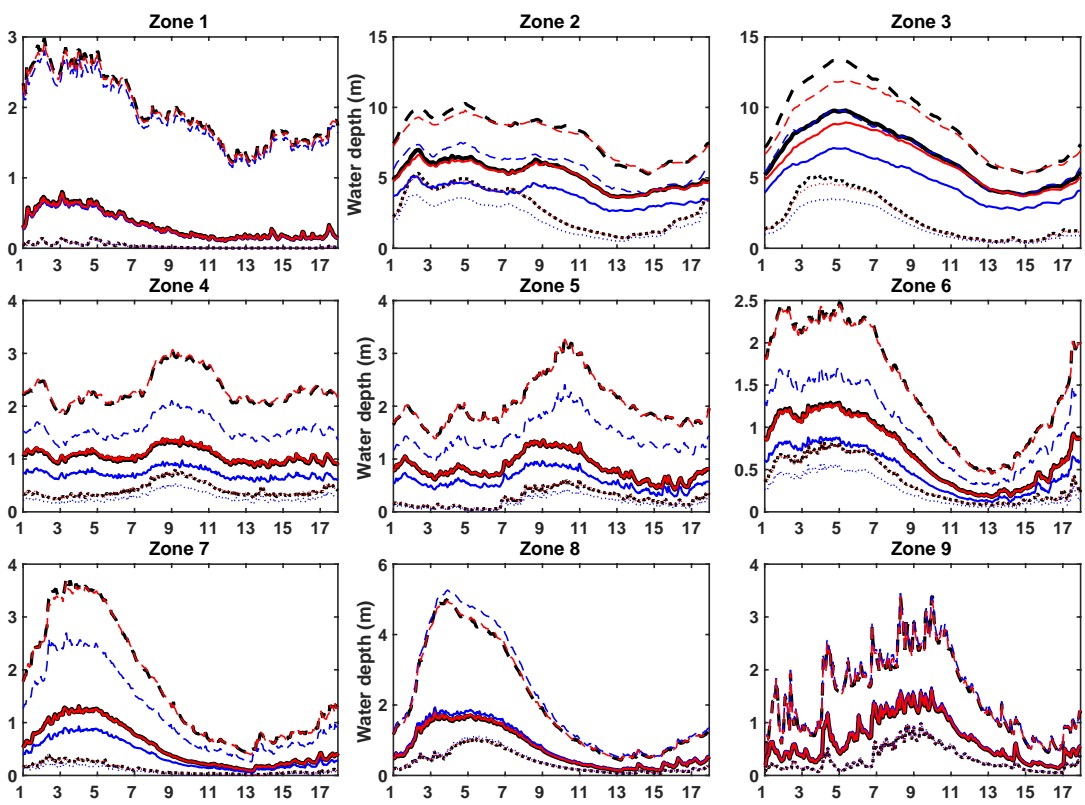

**Figure 10.** Water depths assimilation results for the PE3 experiment: daily evolution of the ensemble-averaged analysis water depth (red lines) compared to the true water depths (black lines) and the openloop water depths (blue line). For each zone (one per subplot), the median (full line), the first decile (dotted line) and the ninth decile (dashed line) of water depth ensemble over all grid cells in the zone are represented.

| Parameter | Nb run | Range |
|:---:|:---:|:---:|
| Spinup | 18 | From 0 window to 17 windows of 21 days |
| Starting date | 17 | Starting January 1st, 2008 and on, every 21 days |
| $n_e$ | 9 | [ 10 20 30 40 50 75 100 150 200 ] |
| $\sigma^b$ | 13 | [ 0.01 0.02 0.05 0.1 0.2 0.3 0.4 0.5 0.75 1.0 1.2 1.5 1.75 ] |

**Table 1.** Tested data assimilation parameters in the sensitivity tests.

| Simulation name | Observation variables | Bathymetry bias |
|:---:|:---:|:---:|
| PE1 | Water depths | No |
| PE2 | Water depth anomalies | No |
| PE3 | Water depth anomalies | Yes |

**Table 2.** List of data assimilation experiments. All experiments are run over approximately one year (17 cycles of 21 days) starting on January 1st, 2008. The ensemble size is $n_e = 25$, the observation error standard deviation is $\sigma^o = 0.1$ m and the initial control variable error standard deviation is $\sigma^b = 0.3$.

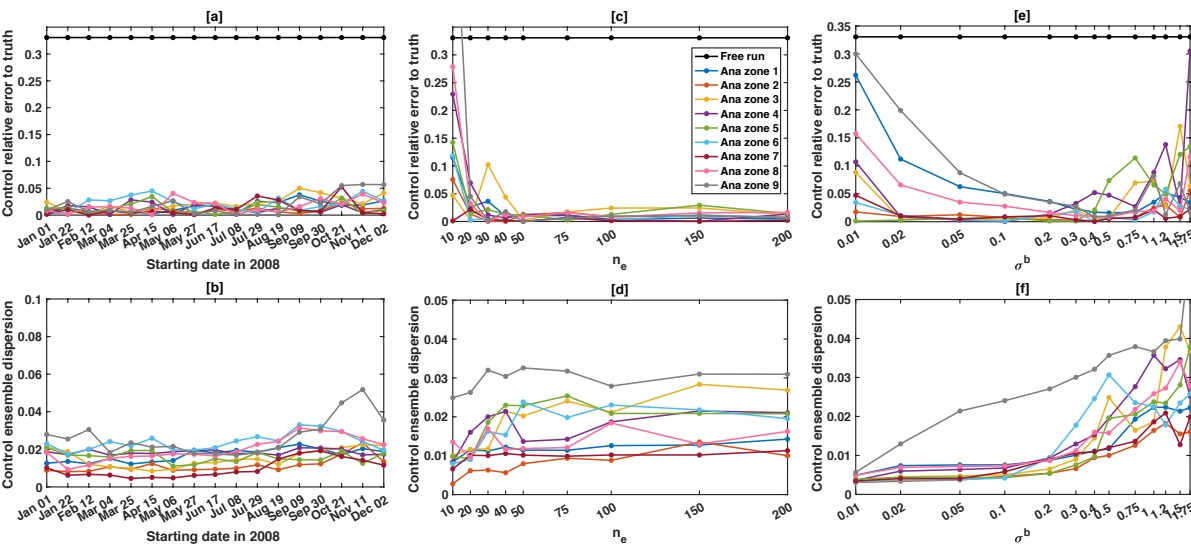

**Figure A1.** (Top) Relative error (to the truth) and (Bottom) dispersion of the analysis control ensemble for each zones for the sensitivity tests to (a-b) the data assimilation starting date, (c-d) the ensemble size $n_e$, (e-f) the background error standard deviation $\sigma^b$. For each test, a set of one-cycle-long data assimilation experiments is run. (Top only) The relative error in zone 1 (dark blue line), zone 2 (orange line), zone 3 (yellow line), zone 4 (purple line), zone 5 (green line), zone 6 (light blue line), zone 7 (burgundy red line), zone 8 (pink line) and zone 9 (gray line) are compared to the basin-averaged openloop relative error (black line).

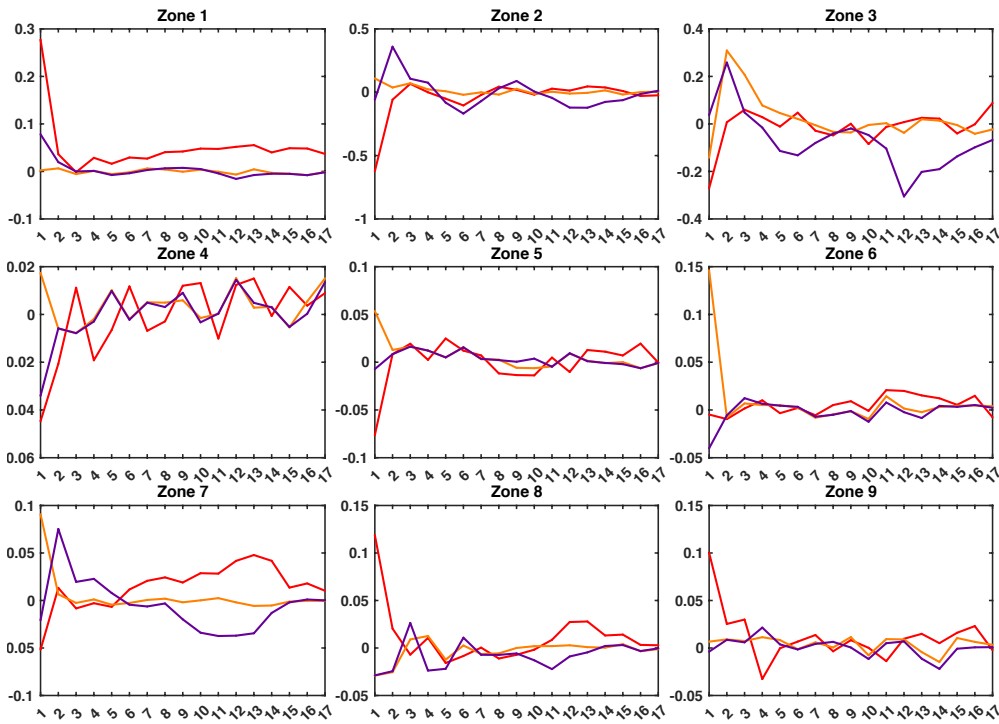

**Figure A2.** Evolution of the EnKF innovations ("$\left( \mathbf{y}_k^{o,[l]} - \mathcal{H}_k(\mathbf{x}_k^{b,[l]}) \right)$" term in Eq. 13) with respect to the assimilation cycle for PE1 (red line), PE2 (orange line) and PE3 (purple line). For each zone (one zone per subplot), the displayed innovation is the averaged of all the innovations in the corresponding zones.

| Run | Starting date | Manning distr. change | Spinup length (in wdws of 21 days) |
|---|---|---|---|
| Reference | Dec 16, 2006 | - | 18 (=378 days) |
| 1 | Dec 16, 2006 | Jan 09, 2007 | 17 (=357 days) |
| 2 | Dec 16, 2006 | Jan 30, 2007 | 16 (=336 days) |
| 3 | Dec 16, 2006 | Feb 20, 2007 | 15 (=315 days) |
| 4 | Dec 16, 2006 | Mar 13, 2007 | 14 (=294 days) |
| 5 | Dec 16, 2006 | Apr 03, 2007 | 13 (=273 days) |
| 6 | Dec 16, 2006 | Apr 24, 2007 | 12 (=252 days) |
| 7 | Dec 16, 2006 | May 15, 2007 | 11 (=231 days) |
| 8 | Dec 16, 2006 | Jun 05, 2007 | 10 (=210 days) |
| 9 | Dec 16, 2006 | Jun 26, 2007 | 9 (=189 days) |
| 10 | Dec 16, 2006 | Jul 17, 2007 | 8 (=168 days) |
| 11 | Dec 16, 2006 | Aug 07, 2007 | 7 (=147 days) |
| 12 | Dec 16, 2006 | Aug 28, 2007 | 6 (=126 days) |
| 13 | Dec 16, 2006 | Sept 18, 2007 | 5 (=105 days) |
| 14 | Dec 16, 2006 | Oct 09, 2007 | 4 (=84 days) |
| 15 | Dec 16, 2006 | Oct 30, 2007 | 3 (=63 days) |
| 16 | Dec 16, 2006 | Nov 20, 2007 | 2 (=42 days) |
| 17 | Dec 16, 2006 | Dec 11, 2007 | 1 (=21 days) |
| 18 | Dec 16, 2006 | Jan 01, 2008 | 0 (=0 days) |

**Table A1.** Spinup sensitivity test set up: each run consists in an approximately-two-year-long ISBA-CTRIP run starting on Dec 16th, 2006 (column 2) and ending on Dec 22nd, 2008. After a given number of 21-day windows during the first year (column 3), the Manning distribution is changed to replicate an assimilation update step while the reference run (row 2) used the same Manning for the entire run. The period between the instant when the Manning distribution is changed and the beginning of the second year of simulation corresponds to the spinup period (column 4). The simulated water depth/discharge during the second year of run are then compared to the reference run in order to evaluate the impact of the spin up.

| Zones | 1 | 2 | 3 | 4 | 5 | 6 | 7 | 8 | 9 |
|---|---|---|---|---|---|---|---|---|---|
| Openloop | 7.57 | 28.84 | 30.28 | 33.51 | 33.41 | 37.15 | 38.83 | 11.48 | 5.23 |
| PE1 | 0.48 | 0.27 | 0.37 | 0.34 | 0.53 | 0.88 | 0.76 | 1.11 | 1.58 |
| PE2 | 1.34 | 0.30 | 1.12 | 0.90 | 0.31 | 1.49 | 0.91 | 0.06 | 1.40 |
| PE3 | 2.23 | 3.52 | 8.31 | 1.48 | 0.57 | 1.39 | 1.56 | 1.67 | 1.46 |

**Table A2.** Zone-averaged RMSEn for the openloop water depths (row 2) and the ensemble-averaged analysis water depths (rows 3-5) compared to the true water depths.

| Zones | 1 | 2 | 3 | 4 | 5 | 6 | 7 | 8 | 9 |
|---|---|---|---|---|---|---|---|---|---|
| Openloop | 2.73 | 4.65 | 6.46 | 3.89 | 5.57 | 4.63 | 9.46 | 3.50 | 3.26 |
| PE1 | 0.35 | 0.38 | 0.34 | 0.15 | 0.25 | 0.53 | 0.29 | 0.30 | 0.59 |
| PE2 | 0.74 | 0.14 | 0.22 | 0.08 | 0.04 | 0.34 | 0.85 | 0.12 | 0.45 |
| PE3 | 1.20 | 4.52 | 7.52 | 0.15 | 0.30 | 0.35 | 2.21 | 1.65 | 2.14 |

**Table A3.** Zone-averaged RMSEn for the openloop discharges (row 2) and the ensemble-averaged analysis discharges (rows 3-5) compared to the true discharges.