# Peer review of "Assimilation of wide-swath altimetry water elevation anomalies to correct large-scale river routing model parameters"

_Hydrology and Earth System Sciences, 2019_

## Referee Comment (RC1) · Hessel Winsemius (Referee) · 25 Jun 2019

Emery et al. propose a new data assimilation scheme (AEnKF), to be used for assimilation of wide-swath altimeter information from the upcoming Surface Water and Ocean Topography (SWOT) mission. Given the large-scale nature and long-time scale revisit times, using an asynchronous scheme, that to the best degree possible utilizes time and space varying availability of information seems like a logical and useful choice, compared to more synchronous approaches such as classical EnKF. I consider this to be a useful contribution to HESS and very much in scope, and useful as a preparation for using SWOT in large-scale hydraulic simulations and forecasts.

[Figure]

I do have a number of comments that lead to my verdict that this paper requires major revisions.

My largest comments are a) the choice to update Manning's n (rather than a state); and related to this, b) the choice to only evaluate the assimilation performance on the basis of water levels (or depth). In most applications, the user will require a good estimate of the river flow (besides water levels), because river flow (a volume in time) controls availability of water for some process that is to be predicted by the model used, not just the water level. A hydrology-hydraulic model-cascade could for instance be used to provide inputs to water allocation predictions for the forthcoming weeks/months (requiring an amount of flow over a given time span), or an upstream boundary condition for a flood simulation of a downstream river stretch. For all such applications, an accurate volume per unit of time is required, not just a stage (except for a flood simulation in a steep area, where floodplain storage is negligible to event accumulated flow, but these are generally small streams, definitely not comparable with the size of the Amazon and its tributaries). I consider this a large (and unnecessary) weakness in the approach, with the additional risk that the Manning roughness will change to physically highly illogical values (which it in fact does in this study!), because e.g. either the water balance of the underlying hydrological simulations of ISBA are biased, or the channel dimensions are poorly defined. To become useful for typical applications, data assimilation should be as much as possible aimed at correcting the amount of water in channel sections, so that predictions after state updating can be made useful and reliable. This is now not proven. I find it a pity that the authors decided to apply this AEnKF on parameters with (As far as I can find in the text) the sole reason being that other authors already used it for state estimation experiments. This makes the study purely theoretical, as I don't really see how the experiments would ever be applied in a real-world case. The authors should at least show river flow as an additional benchmark variable and show how the 3 experiments affect the accuracy of river flow and discuss this result. My logical feeling is that discharge will be quite heavily impacted especially in experiment 3 where a bias is introduced.

Second point: I don't fully understand the zonal approach to updating Manning's n. To me it would make more sense to use an upstream-downstream relation in manning coefficients (e.g. update manning coefficient at location, as well as upstream and downstream) which could easily introduce a logical covariance between n values (rather than assuming everything to be independent). In fact, the full zonal approach with areas that may have very little relationship to each other suggests, that there is a 100% covariance across the zone. Why was this selected in this way (or would things become overcomplicated if done in a different way)? Consider discussing this in the last sections of the paper.

The English writing and sentence constructions are not everywhere up to standards. Please make sure the manuscript is reviewed by a (near-)native English person.

Detailed comments are listed below:

Introduction: there are many "however"s in the text. Some or many of these can be removed.

P. 3, l. 32 "at a coarser scale", please just describe the scale.

p. 4. l. 13: "gravitational drainage", do you mean groundwater outflow?

p. 4. l. 20. Replace "empties" by "spills"

p. 4, l. 29, Replace "fixes" by "results in"

p. 5 l. 2 (p.5): "...values between 0. 75 and 1 for smaller and mountainous tributaries..." I guess you mean 0.075 and 0.1 s m-1/3. The values you mention are ridiculously high!

p. 5. Eq. 1. Why is SOmax not simply 1 as it is only a way to scale values?

p. 5, l. 10 replace "as the cells approaches" for "towards"

p. 5, l 11. V(t) is not the surface flow, but the average cross-sectional flow velocity.
p. 5, eq. 3. S is not defined

p. 6, l. 3. "forcings are considered perfect". This is my point above. They never are and the assimilation should work to correct these forcings. In the case of ISBA this concerns errors in the water balance, and in CTRIP errors in the transport of mass and momentum through the channel network.

p. 6, l. 26. "white noise", is this a reasonable assumption? And if reality is different, how would it affect your results? Discuss this in Section 7.

p. 11, eq 17 and 18. Are these the selected zonal Manning;s n values? These are unrealistically high. Why are these selected in such a strange domain?

p. 11, 19. Describe briefly what your expected results are (i.e. why these experiments) on both water levels and flows!

Section 5.1. Describe what you I hydrological sense expect from the spinup time experiment. You can relate the expected required spinup to the time of concentration of the considered basin.

p. 13. L. 9, you only show spatial average results. Why not spatial patterns? That may reveal the locations where things go right / wrong.

p. 15, l. 16-28. I find this paragraph not very clear, it is not clear why the results behave so differently from zone 1,2,3. I was wondering if there is not simply too little water in the system to get correct results in this part of the domain? If you only change manning's n, you can never introduce new water or take water out of the system (see main comment).

p. 17, l. 20. Around this part you should definitely discuss the state updating versus parameter updating, and the water storage errors that you can never resolve with parameter updating.

p. 17, l. 31. I am very curious what kind of exceptional hydrological event you mean

here.

Some of the figures have too small fonts, please make the figures readible throughout the text.

---

## Referee Comment (RC2) · Claire I. Michailovsky (Referee) · 1 Jul 2019

**General Comments**

This paper presents a data assimilation experiment using synthetic observations from the planned Surface Water Ocean Topography (SWOT) mission to update the Manning roughness parameters of a large scale routing model. The assimilation method used in the Asynchronous Ensemble Kalman Filter (AEnKF) with a 21-day time window (one orbit repeat cycle) which allows for measurements at different locations acquired at different times to be included in a single assimilation step.

[Figure]

The application of the AEnKF is logical considering the specificities of the data and this is a useful contribution to the work preparing for SWOT.

I recommend publication with major revisions, including a thorough review for improving language.

My main comment relates to the specificity of the study as a synthetic experiment and with how some of the simplifications/assumptions are presented. I recommend further discussion on the impact of these on a study using real data, and on the estimates of uncertainties which are crucial to any assimilation experiment.

These are specifically:

1) The assimilation of depth rather than elevation

2) The fact that the truth is generated by the same model:

- no model structure error

- there is a "real" Manning to converge to, which might not be the case with real data

- even in PE3, the bathymetry is not significantly changed, only the river bed elevation.

3) Assumption of perfect forcing

Specific Comments

P6, l.21: this is brought up later when the anomalies are assimilated, but more focus should be placed here on the fact that depth is assimilated while SWOT will produce elevations. The conversion from level to depth is one of the big issues with using altimetry in hydrological studies. Does the SWOT simulator directly produce depths? You are assuming known river bed elevation, and this should be clearly specified. (I can see this is mentioned p11, l.26 but that is too late in the paper).

P7, l.9: similar issue as previous comment, elevation and not depth is required at 10cm accuracy. The depth accuracy will be much lower. You assume no representativeness

error due to scale, but how about level vs. depth?

P8, l11: Would this be necessary if you had a better representation of your measurement error (re: previous comments)? What is the magnitude of this additional error?

Technical Corrections

At some points vague language is used (f.ex: "without really converging"), please be more precise.

P7, l.25: It is not clearly explained if xk are the Mannings coefficients themselves or the multiplying factors as described above. This issue is repeated elsewhere in the paper and while I understand xk does refer to the factors, please make sure this is clear throughout.

Figure 8: the black lines cannot really be seen, perhaps increase the width?

See attached pdf file for other technical corrections and comments

Please also note the supplement to this comment:
https://www.hydrol-earth-syst-sci-discuss.net/hess-2019-242/hess-2019-242-RC2-supplement.pdf

[Figure]

**Supplement:**

[revised manuscript text omitted]

---

## Referee Comment (RC3) · Paul Bates (Referee) · 25 Jul 2019

In addition to the comments of the previous referees, I would add that I think the paper also needs to better articulate what is the new contribution to knowledge made by the work. Previous papers have established that SWOT data is very likely going to allow us to better constrain parameters and states in a variety of environmental models, but I was missing what in addition to this that we learn from the present work. There are a number of specific technical differences here to the previous papers by Pedinotti et al (2014) and Emery et al (2016) which are described in the last paragraph of the introduction, but I am not sure these changes lead to very different conclusions. Overall

the abstract states that the assimilation scheme " is able to retrieve the true value of the Manning coefficients within one assimilation cycle most of the time", but isn't this ability already well established, at least in general terms? What do we learn about the likely potential of the SWOT data that we did not know before? Alternatively, what do we learn about the kinds of assimilation schemes that are going to work well with the SWOT data? In addressing this issue the comments of the other referees about the rather specific nature of the numerical experiment are pertinent: if the experiment does not reflect the assimilation problem that will be faced in practice when ISBA-CTRIP is confronted with SWOT data then the conclusions drawn may not be easy to generalize. I think a significant effort needs to be made to re-draft the abstract introduction to more clearly identify what is the new contribution to knowledge made by the paper. I'm sure there is one, so this is more a question of properly assembling the arguments in order to bring this out.

―――――――――――――――――

---

## Author Comment (AC1) · 25 Nov 2019

The authors would like to thank referee Paul Bates for his helpful comments. Please find in the attached file " " our replies with which we hope to address his concerns. To easily address all comments, we numbered each comment: they are referenced with a "M" for the major comments and with a "m" for the minor comments.

The abstract, the introduction (the latest paragraphs only) and the discussions sections have been updated a lot due to several remarks from all reviewers. Therefore, we attached to our reply separate files with the new version of these sections. In these rewritten sections, we used a color code to differentiate which reviewer made the com-

ment and suggested a modification: comments from reviewer 1 (Hessel Winsemius), 2 (Claire Michailovsky) and 3 (Paul Bates) are in purple, orange and green respectively. Each modification is also referenced by a code in bracket indicating the reviewer (R#***) and the type and index of the comment (M/m#***) such as: "[R#3-M#1]".

Documents in the attached archive: - Authors' replies: Reply-to-reviewer-3.pdf - Corrected abstract: Hess-2019-242-corrected-abstract.pdf - Corrected introduction (latest paragraphs: From p.3, l.19 – p.4, l.5 in the original version of the manuscript): Hess-2019-242-corrected-introduction.pdf

Please also note the supplement to this comment:
https://www.hydrol-earth-syst-sci-discuss.net/hess-2019-242/hess-2019-242-AC1-supplement.zip

---

## Author Comment (AC2) · 25 Nov 2019

The authors would like to thank referee Claire Michailovsky for her detailed and very helpful comments. She raises very pertinent points that helped us better articulate our purpose. Please find attached our replies and associated modifications to the manuscript with which we hope to address her concerns. To easily address all comments, we numbered each comment: they are referenced with a "M" for the major comments and with a "m" for the minor comments.

We kept the reviewer's comments and question in bold while our replies are in italic. When the associated manuscript's modifications remain small, we inserted the modified paragraph in our reply in plain text: the black text corresponds to the original unmodified text, the crossed text corresponds to deleted text and the blue text corresponds to new text.

The abstract, the introduction (the latest paragraphs) and the discussions sections have been more updated due to several remarks from all reviewers. Therefore, we attached to our reply separate files with the new version of these sections. In these rewritten sections, we used a color code to differentiate which reviewer made the comment and suggested a modification: comments from reviewer 1 (Hessel Winsemius), 2 (Claire Michailovsky) and 3 (Paul Bates) are in purple, orange and green respectively. Each modification is also referenced by a code in bracket indicating the reviewer (R#***) and the type and index of the comment (M/m#***) such as: "[R#3-M#1]".

Documents in attached archives: - Replies to reviewer: Reply-to-reviewer-2.pdf - Modified abstract: Hess-2019-242-corrected-abstract.pdf - Modified introduction (latest paragraphs: from p.3, l.19 – p.4, l.5 in original version of the manuscript): Hess-2019-242-corrected-introduction.pdf - Modified discussions: Hess-2019-242-corrected-discussions.pdf

Please also note the supplement to this comment:
https://www.hydrol-earth-syst-sci-discuss.net/hess-2019-242/hess-2019-242-AC2-supplement.zip
* * *
**Zone 1**

**Zone 2**

**Zone 3**

**Zone 4**

**Zone 5**

**Zone 6**

**Zone 7**

**Zone 8**

**Zone 9**

Water depth (m)

**Fig. 1.** Modified Figure 8 to address comment m#6

[Figure]

Fig. 2. Modified Figure 10 to address comment m#6

---

## Author Comment (AC3) · 25 Nov 2019

The authors would like to thank referee Hessel Winsemius for his detailed and very helpful comments, which helped us to better articulate our paper. Please find attached our replies and associated modifications to the manuscript in order to address these concerns. To easily address all comments, we numbered each comment: they are referenced by a "M" for the major comments or by a "m" for the minor comments.

We kept the reviewer's comments and question in bold while our replies are in italic. When the associated modifications in the manuscript remain small, we inserted the modified paragraph in our reply in plain text: the black text corresponds to the original

unmodified text, the crossed text corresponds to deleted text and the blue text corresponds to new text.

The abstract, the introduction (the latest paragraphs) and the discussions sections have been heavily modified to take into account several remarks from the three reviewers. Therefore, we attached to our reply separate files with the new version of these three sections. In these rewritten sections, we used a color code to differentiate which reviewer made the comment and suggested a modification: comments from reviewer 1 (Hessel Winsemius), 2 (Claire Michailovsky) and 3 (Paul Bates) are in purple, orange and green respectively. Each modification is also referenced by a code in bracket indicating the reviewer (R#***) and the type and index of the comment (M/m#***) such as: "[R#3-M#1]".

Documents in attached archive: - Reply to reviewer: Reply-to-reviewer-1.pdf - Modified abstract: Hess-2019-242-corrected-abstract.pdf - Modified introduction (the latest paragraphs - from p.3, l.19 – p.4, l.5 in original version of the manuscript) : Hess-2019-242-corrected-introduction.pdf - Modified discussions: Hess-2019-242-corrected-discussions.pdf

Please also note the supplement to this comment:
https://www.hydrol-earth-syst-sci-discuss.net/hess-2019-242/hess-2019-242-AC3-supplement.zip
* * *
**Fig. 1.** Modified Figure 7 to address comment m#20

**Fig. 2.** Modified Figure 8 to address comment m#20

**Fig. 3.** Modified Figure 9 to address comment m#20

Zone 1

Zone 2

Zone 3

Water depth (m)

Zone 4

Zone 5

Zone 6

Water depth (m)

Zone 7

Zone 8

Zone 9

Water depth (m)

**Fig. 4.** Modified Figure 10 to address comment m#20

[Figure]

**Fig. 5.** Additional Figure in Appendix: same as Figure 6 but per zone to address comment m#16

---

## Author Response (AR1)

**Emery et al. propose a new data assimilation scheme (AEnKF), to be used for assimilation of wide-swath altimeter information from the upcoming Surface Water and Ocean Topography (SWOT) mission. Given the large-scale nature and long-time scale revisit times, using an asynchronous scheme, that to the best degree possible utilizes time and space varying availability of information seems like a logical and useful choice, compared to more synchronous approaches such as classical EnKF. I consider this to be a useful contribution to HESS and very much in scope, and useful as a preparation for using SWOT in large-scale hydraulic simulations and forecasts.**

*The authors would like to thank referee Hessel Winsemius for his detailed and very helpful comments, which helped us to better articulate our paper. Please find below our replies and associated modifications to the manuscript in order to address these concerns. To easily address all comments, we numbered each comment: they are referenced by a "M" for the major comments or by a "m" for the minor comments.*

*We kept the reviewer's comments and question in bold while our replies are in italic. When the associated modifications in the manuscript remain small, we inserted the modified paragraph in our reply in plain text: the black text corresponds to the original unmodified text, the crossed text corresponds to deleted text and the blue text corresponds to new text.*

*The abstract, the introduction (the latest paragraphs) and the discussions sections have been heavily modified to take into account several remarks from the three reviewers. Therefore, we refer the reviewers directly to the corrected annotated manuscript. In the annotated manuscript, we used a color code to differentiate which reviewer made the comment and suggested a modification: comments from reviewer 1 (Hessel Winsemius), 2 (Claire Michailovsky) and 3 (Paul Bates) are in purple, orange and green respectively. Each modification is also referenced by a code in bracket indicating the reviewer (R#***) and the type and index of the comment (M/m#***) such as: "[R#3-M#1]".*

**I do have a number of comments that lead to my verdict that this paper requires major revisions.**

**MAJOR COMMENTS**

**M#1. My largest comments are a) the choice to update Manning's n (rather than a state); and related to this, b) the choice to only evaluate the assimilation performance on the basis of water levels (or depth). In most applications, the user will require a good estimate of the river flow (besides water levels), because river flow (a volume in time) controls availability of water for some process that is to be predicted by the model used, not just the water level. A hydrology-hydraulic model-cascade could for instance be used to provide inputs to water allocation predictions for the forthcoming weeks/months (requiring an amount of flow over a given time span), or an upstream boundary condition for a flood simulation of a downstream river stretch. For all such applications, an accurate volume per unit of time is required, not just a stage (except for a flood simulation in a**

**steep area, where floodplain storage is negligible to event accumulated flow, but these are generally small streams, definitely not comparable with the size of the Amazon and its tributaries). I consider this a large (and unnecessary) weakness in the approach, with the additional risk that the Manning roughness will change to physically highly illogical values (which it in fact does in this study!), because e.g. either the water balance of the underlying hydrological simulations of ISBA are biased, or the channel dimensions are poorly defined. To become useful for typical applications, data assimilation should be as much as possible aimed at correcting the amount of water in channel sections, so that predictions after state updating can be made useful and reliable. This is now not proven. I find it a pity that the authors decided to apply this AEnKF on parameters with (As far as I can find in the text) the sole reason being that other authors already used it for state estimation experiments. This makes the study purely theoretical, as I don't really see how the experiments would ever be applied in a real-world case. The authors should at least show river flow as an additional benchmark variable and show how the 3 experiments affect the accuracy of river flow and discuss this result. My logical feeling is that discharge will be quite heavily impacted especially in experiment 3 where a bias is introduced.**

*We agree with the reviewer on the fact that river flow/discharge is one of the main variable of interest for river modeling applications and, following that, state estimation are well adapted for hydrology data assimilation applications. However, we would like to emphasize that we already dedicated another study focused on the correction of the state and the discharge in the context of remotely-sensed hydrology products, see Emery et al. (2018). Moreover, as SWOT is a scientific mission with a nominal lifespan of 3 years (we indicate now this point in the abstract), an important application of SWOT data would be to calibrate hydrological models in order to get better simulations over past and future periods. Therefore, the present parameter estimation study works as a complementary study focusing on correcting crucial parameters for hydrological applications that are still not well known. To set this framework clearer, we added those remarks at the end of the introduction. Still, it is possible to apply the current framework (e.g. twin experiments with SWOT-like observations) to correct either water depths or river discharges (state estimation). We added this point to the perspectives and future works in the manuscript's conclusions.*

*The choice of the Manning coefficient as control variable is directly linked to the results of a sensitivity analysis (SA) of the ISBA-CTRIP river outputs to its routing parameters (Emery et al 2016). This SA showed that, among all CTRIP parameters and their tested ranges, simulated water depths are essentially sensitive to the Manning coefficients. Then, as SWOT water elevations product is closest to ISBA-CTRIP's water depths, it was chosen to build a framework based on the assimilation of SWOT-like water depths to correct the model most sensitive input parameters.*

*Besides, the reviewer also made very relevant points regarding the consequence of forcing/LSM bias on the roughness coefficient value. Indeed, in a real-case framework, if these types of bias are not considered, the assimilation scheme will try to change Manning coefficient values to compensate water elevations variations due to these biases. We realized that this point was not discussed in the initial manuscript, so we dedicated a new paragraph in the Discussions section where we acknowledge this limitation of our current approach and suggest solutions to handle it in future developments.*

*Moreover, we would like to clarify that it is not the direct value of the Manning coefficient that is corrected but a multiplying factor applied to the Manning distribution (see our reply to your M#2). Therefore, the high values for the control variable displayed in Figures 7 and 9 correspond to these multiplying factors which remain within a range of 0.5 and 1.5, while the corresponding Manning coefficients are within physical value between 0.02 and 0.07.*

*Finally, as suggested by the reviewer, the model and data assimilation performances on water depth and discharge were easily calculated. Given the idealized framework of the OSSE, the statistics were exceptionally good for both water depth and discharge and did not bring essential new information to the manuscript. These tables were then added to the manuscript in a dedicated appendix.*

- *Modified introduction >> see annotated manuscript P.3 L.29-33 + P.3 L.35 – P.4 L.4 + P.4 L.11-14*
- *Modified manuscript in the study's perspectives, P.21 L.32, L.35 – P.22 L.1-3:*

These experiments offer several perspectives. They mainly consist in going towards more realistic data assimilation experiments that take into account more sources of uncertainties between the model and the observations, {such as correlated observation errors or uncertainties in the forcing and LSM surface and sub-surface runoff}. To test the performances' limitations regarding the DEM/bathymetry bias issue, one can use simulated water surface elevations referenced to a geoid instead of water depths from the model or even assimilate water depths from another model where the bathymetry is different. As most applications generally require a good estimate of the river flow and river water volume, another lead of investigation could maintain the SWOT-based OSSE framework but to correct the simulated water storage and/or discharge, either as a single state estimation framework or as a dual state-parameter estimation framework (similarly to dual discharge-bathymetry inference methods developed by Oubanas et al., 2018 and Brisset et al. 2018 for some hydraulic models). Moreover, along with observations of water surface elevations, SWOT will also provide two-dimensional maps of river widths and surface slopes. One can also study the possibility of assimilating such product to correct the corresponding parameters in ISBA-CTRIP such as the model river width or maybe constrain other parameters such as the bankful depth that controls the model flooding scheme.

- *Modified discussions >> see annotated manuscript P.19 L.24-29*
- *Added appendix D*

**M#2. Second point: I don't fully understand the zonal approach to updating Manning's n. To me it would make more sense to use an upstream-downstream relation in manning coefficients (e.g. update manning coefficient at location, as well as upstream and downstream) which could easily introduce a logical covariance between n values (rather than assuming everything to be independent). In fact, the full zonal approach with areas that may have very little relationship to each other suggests, that there is a 100% covariance across the zone. Why was this selected in this way (or would things become overcomplicated if done in a different way)? Consider discussing this in the last sections of the paper.**

*The zonal approach is applied to the control variables only. As previously mentioned in the reply to M#1, the control variablse are not directly the Manning coefficients but rather a set of multiplying factors applied to the Manning coefficient distribution. It is those factors that are set constant over the 9 hydro-geomorphologic areas while the Manning coefficient distribution remains spatially-distributed at the grid-cell scale and built on an upstream-downstream relationship (specified in Eq. 1 of the manuscript). The interest of using such zonal approach to correct the Manning coefficient distribution was indeed to maintain this upstream-downstream relationship between the grid-cell (which was not the case when each Manning coefficient was individually updated in Pedinotti et al.*

*2014). We realized that this aspect might not be clear enough in the manuscript as reviewer #2 also raised similar questions. Therefore, we modified the manuscript adequately in Section 3.2.2 to better present the definition of the control variable and recall it in Sections 4 and 6.*

- *Modified manuscript in section 3.2.2: P.8 L.18-24*

Following the conclusions from the ISBA-CTRIP sensitivity analysis to its routing parameters in Emery et al. (2016), we determined that assimilating water-depth-like observations would be efficient to correct the distribution of the river Manning coefficients. These coefficients are spatially-distributed at the grid-cell scale. However, from Pedinotti et al. (2014), equifinality issues were raised by correcting the distribution at this scale while also affecting its upstream-to-downstream spatial distribution. Thus, we chose to correct it by applying multiplying factors defined at a coarser scale, namely at the scale of the 9 hydro-geomorphological areas defined in Section 2.3 and illustrated in Figure 1b. Within a same area, the Manning coefficients of all grid-cells are identically updated by being multiplied by the same correcting factor. Thus, data assimilation will focus on directly adjusting these multiplying factors. Therefore, the control vector is composed of the nx multiplying factors $N_{mult;i}$, i = 1: : :$n_x$, applied to correct the distribution of the river Manning coefficient:

$$x_k = \left[N_{mult,1}, \ldots, N_{mult,n_x}\right]^T, (8)$$

giving nx = 9.

- *Modified manuscript in section 4: P.12 L.17, L.19*

In the incoming experiments, the true control variables xt are:
…. (17)
and their a priori values at the first assimilation cycle xb are:
…  (18)

- *Modified manuscript in section 6: P.16 L.12-13*

We present now the results from the data assimilation experiments presented in Table 2 and in Section 4.4.2. Recall that these experiments aim at correcting a set of 9 multiplying factors applied to the manning coefficient distribution and constant over 9 hydro-geomorphological zones  covering the Amazon basin.

**M#3. The English writing and sentence constructions are not everywhere up to standards. Please make sure the manuscript is reviewed by a (near-)native English person.**

*Reviewer #2 had a similar remark. Therefore, while preparing the replies to the reviewers, we submitted the manuscript to an independent English-speaking proofreader to improve the English.*

**MINOR COMMENTS**

**m#1. Introduction: there are many "however"s in the text. Some or many of these can be removed.**

*Thank you for noticing this. We read through the introduction and modified it to use different linking words.*

**m#2. P. 3, l. 32 "at a coarser scale", please just describe the scale.**

*Here, the "coarser resolution" relates to the zonal distribution, in opposition to the grid-cell finer resolution. As the zonal distribution is not introduced yet in the manuscript, it is true that it might be confusing. Therefore, we modified the sentence in the introduction to be more explicit.*

- *Modified manuscript in Introduction P.4 L.22-23*

For the current study, it was decided to update the Manning coefficient distribution, not a the grid-cell resolution but at a coarser zonal resolution, by applying multiplying correcting factors constant over each zones,  identical to the one used in Emery et al. (2016).

**m#3. p. 4. l. 13: "gravitational drainage", do you mean groundwater outflow?**

*Not exactly. When we use the term "gravitational drainage", we are describing the LSM where the water flows toward the deep soil and feeds CTRIP's groundwater reservoir (denoted G). The term "groundwater outflow" is used for CTRIP and represents the flow from the groundwater reservoir G into CTRIP's river/surface reservoir S. It is now specified in the manuscript.*

- *Modified manuscript in section 2.1: P.5 L.8*

In particular, ISBA gives a diagnostic of the surface runoff (QISBA,sur) and the gravitational drainage (QISBA,sub, , i.e. water percolating to the deep layers of the soil ) later used as forcing inputs for the RRM denoted CTRIP.

**m#4. p. 4. l. 20. Replace "empties" by "spills"**

*Thank you this suggestion. The modification was made into the manuscript.*

**m#5. p. 4, l. 29, Replace "fixes" by "results in"**

*Thank you this suggestion. The modification was made into the manuscript.*

**m#6. p. 5 l. 2 (p.5): ". . .values between 0. 75 and 1 for smaller and mountainous tributaries. . ." I guess you mean 0.075 and 0.1 s m-1/3. The values you mention are ridiculously high!**

*Yes, you are absolutely right. This is a typing error, thank you for noticing it. The manuscript has been corrected.*

**m#7. p. 5. Eq. 1. Why is SOmax not simply 1 as it is only a way to scale values?**

*Actually, there is another typing error here. SO is the stream order taking values ranging from 1 at source cells (grid cells without any upstream grid cells, according to the river network) to a maximal SO associated to the outlet grid cell (depending on the depth of the river network). It is Nmin and Nmax that takes values of 0.04 and 0.06, following the configuration from Decharme et al (2012). These definitions were corrected in the manuscript.*

- *Modified manuscript in section 2.2: P.6 L.3-6*

 SO  the stream size relative measure at the current cell; SOmax =  is the same measure at the river mouth (which value depends on the depth of the river network) and SOmin =  the measure at source cells (namely cells without any upstream cells according to the river network) . The Manning coefficient is then set to be constant in time while its spatial values decrease as the cells approaches the river outlet, taking values between Nmin=0.04 and Nmax=0.06 (Decharme et al 2012). .

**m#8. p. 5, l. 10 replace "as the cells approaches" for "towards"**

*Thank you this suggestion. The modification was made into the manuscript.*

**m#9. p. 5, l 11. V(t) is not the surface flow, but the average cross-sectional flow velocity.**

*You are right. By "surface", we meant the flow velocity in CTRIP's surface reservoir S. The confusion is now corrected in the manuscript.*

- *Modified manuscript in section 2.2: P.6 L.7-8*

All these parameters are eventually essential to estimate the spatially- and time-varying  average cross-sectional flow velocity in the surface reservoir v(t) following the Manning formula.

**m#10. p. 5, eq. 3. S is not defined**

*Actually, it was introduced p4, l.17, but it is true that it is hidden within the text and does not clearly appear as a variable. Therefore, we added it in p.5*

- *Modified manuscript in section 2.2: P.6 L.10*

where hS is the river water depth estimated from the river storage S by

**m#11. p. 6, l. 3. "forcings are considered perfect". This is my point above. They never are and the assimilation should work to correct these forcings. In the case of ISBA this concerns errors in the water balance, and in CTRIP errors in the transport of mass and momentum through the channel network.**

*We agree with the reviewer that the forcings are never perfect. By "perfect" here, we meant that the forcing uncertainties are not included in the generation of the ensemble, but they should definitely be included in future studies. First, in the manuscript, we withdraw the use of the term "perfect" when presenting the forcing (section 2.4) but instead write it is considered "as such" in the data assimilation framework (section 3.4.3). Then, following a similar remark from reviewer #2, we dedicate a paragraph to the "perfect" forcing assumption in the discussions. Finally, the objective to directly correct these forcings with assimilation is here out of the scope of the study but it is pointed out in the discussions.*

- *Modified manuscript in section 2.4.*

- *Modified manuscript in section 3.4.3: P.11 L.14-17*

To generate the background control ensemble, we solely stochastically perturb the variables within the control vector. Note that, by only perturbing the variables in the control vectors to generate the ensemble, we assume that all other features of the forward model, e.g. the atmospheric forcings, the LSM structure and therefore the surface and sub-surface runoff, are perfect. While this is the case for purely OSSE, such features are never perfect in real-case experiment. This assumption is further discussed in the Section 6.

- *Modified discussions P.19 L.21-24*

**m#12. p. 6, l. 26. "white noise", is this a reasonable assumption? And if reality is different, how would it affect your results? Discuss this in Section 7.**

*When the study was developed, there was, at the time, no large-scale SWOT simulator allowing considering correlated SWOT-like errors so the white noise assumption was the most reasonable, in default of having a better error model. Nevertheless, reviewer #2 had a similar comment inviting us to discuss more those observation errors in the Discussions; therefore we added a dedicated paragraph in the Discussions.*

- *Modified discussions P.20 L.17-27*

**m#13. p. 11, eq 17 and 18. Are these the selected zonal Manning;s n values? These are unrealistically high. Why are these selected in such a strange domain?**

*No, those are the multiplying factors of the Manning coefficient distribution. This explains why their values are around 1.0. Following the modifications from M#2, this is now explicitly specified in the Section.*

**m#14. p. 11, 19. Describe briefly what your expected results are (i.e. why these experiments) on both water levels and flows!**

*Following the reviewer's suggestion, we added a few sentences in the Section 4.2.*

- *Modified manuscript in section 4.2: P.13 L.8-10*

The first experiment, denoted as PE1, is configured from the aforementionned sensitivity test outcome. The parameters defining the experiment (spinup, starting date, ensemble size, control error) will be those giving the best results in the sensitivity tests in Table 2. Also, the reference level between the observed and simulated water depths is the same. In other words, there is no bias in the observation. This first idealized experiment serves as proof-of-concept as the observations nature matches exactly the type of the simulated variables. Consequently, with this experiment, we expect to retrieve the true value of the control variables and hence the correct water depths and discharges.

- *Modified manuscript in section 4.2: P.13 L.26-30*

First, in experiment PE2, there will still be no bias between the observeo and simulated river bathymetry to see how the assimilation of anomalies performs. Similarly to PE1, we expect this experiment to be able to retrieve the true control and state variables. Finally, the last experiment PE3, which introduces a constant relative bias between CTRIP and SWOT, will be carried out. For this experiment, we anticipate that the assimilation will still be able to retrieve the model state variables. The use of anomalies as observations should limit the impact of the inserted bias however, we do not exclude that it may be slightly echoed on the control variables.

**m#15.** Section 5.1. Describe what you I hydrological sense expect from the spinup time experiment. You can relate the expected required spinup to the time of concentration of the considered basin.

*Thank you for this suggestion. We added the following sentences at the end of section 5.1.*

- *Additional remark in section 5.1: P.14 L.26-27*

This period corresponds to the basin concentration time or, in other terms, the required time for the river network to totally empty.

**m#16.** p. 13. L. 9, you only show spatial average results. Why not spatial patterns? That may reveal the locations where things go right / wrong.

*You are right. Initially, we chose to display the results averaged over the basin to keep the figure easy to read and limit the number of figures within the manuscript. However, these plots can definitively be generated. Therefore, we added the corresponding figure in a new appendix.*

- *Added Appendix C*

**m#17.** p. 15, l. 16-28. I find this paragraph not very clear, it is not clear why the results behave so differently from zone 1,2,3. I was wondering if there is not simply too little water in the system to get correct results in this part of the domain? If you only change manning's n, you can never introduce new water or take water out of the system (see main comment).

*You are right. We focused on the sensitivity analysis results to explain our results but it is true that, during the low flow period, there might be just too little water entering the system and the assimilation is unable to perform efficiently probably due to a forecast ensemble that is not spread enough. The section was slightly modified to include this comment and to simplify the interpretation.*

- *Modified manuscript in section 6.1: P.17 P.12-15*

Subsequently, zones 6, 7 and 8 correspond to right-bank tributaries, namely the Juruá and Purus rivers (zone 6), the Madeira river (zone 7) and the Tapajós and Xingu rivers (zone 8). These right-bank tributary zones are characterized by a strong seasonal cycle (see Figure 8, zones 6-8). Then, by comparing the corresponding plots in Figures 7 and 8, we notice that the period when the analysis control variable spreads from the truth corresponds to the low flow season in these zones. According to the global sensitivity analysis results, water depths in these zones are less sensitive to the Manning coefficient in low flow conditions. Additionally, there is very little water in the zones during this period. Consequently, the background control ensemble is not spread enough for the EnKF to be efficient. Meanwhile, the EnKF still "sees"  that the observations are higher than the model predictions (as seen with the positive innovation in these zones shown  in Figure A1).Therefore, in order to increase the simulated water depths, the EnKF corrects the Manning coefficient so that its value gets higher (a higher Manning coefficient means a slower flow velocity and then a higher simulated water depth).  Finally, once the low flow season ends, the analysis Manning coefficient converges back to the truth (see the last assimilation cycles).

**m#18.** **p. 17, l. 20. Around this part you should definitely discuss the state updating versus parameter updating, and the water storage errors that you can never resolve with parameter updating.**

*Following the reviewer's suggestion, we completed the Discussions paragraph on the forcing and LSM bias.*

- *Modified discussions P.19 L.24-29*

**m#19.** **p. 17, l. 31. I am very curious what kind of exceptional hydrological event you mean here**

*Maybe the term was not the most adapted here. By "exceptional", we meant intense flooding for example. We also changed the term by "extreme".*

- *Modified discussions P.18 L.29-30*

**m#20.** **Some of the figures have too small fonts, please make the figures readible throughout the text.**

*Following the reviewer's suggestion and a similar comment from reviewer #2, we re-worked the figures, namely Figures 7 to 10.*

**Hess-2019-242-RC2**
**Reply to Claire Michailovsky's comments**

**General Comments**

**This paper presents a data assimilation experiment using synthetic observations from the planned Surface Water Ocean Topography (SWOT) mission to update the Manning roughness parameters of a large scale routing model. The assimilation method used in the Asynchronous Ensemble Kalman Filter (AEnKF) with a 21-day time window (one orbit repeat cycle) which allows for measurements at different locations acquired at different times to be included in a single assimilation step. The application of the AEnKF is logical considering the specificities of the data and this is a useful contribution to the work preparing for SWOT.**

*The authors would like to thank referee Claire Michailovsky for her detailed and very helpful comments. She raises very pertinent points that helped us better articulate our purpose. Please find below our replies and associated modifications to the manuscript with which we hope to address her concerns. To easily address all comments, we numbered each comment: they are referenced with a "M" for the major comments and with a "m" for the minor comments.*

*We kept the reviewer's comments and question in bold while our replies are in italic. When the associated manuscript's modifications remain small, we inserted the modified paragraph in our reply in plain text: the black text corresponds to the original unmodified text, the crossed text corresponds to deleted text and the blue text corresponds to new text.*

*The abstract, the introduction (the latest paragraphs) and the discussions sections have been heavily modified to take into account several remarks from the three reviewers. Therefore, we refer the reviewers directly to the corrected annotated manuscript. In the annotated manuscript, we used a color code to differentiate which reviewer made the comment and suggested a modification: comments from reviewer 1 (Hessel Winsemius), 2 (Claire Michailovsky) and 3 (Paul Bates) are in purple, orange and green respectively. Each modification is also referenced by a code in bracket indicating the reviewer (R#\*\*\*) and the type and index of the comment (M/m#\*\*\*) such as: "[R#3-M#1]".*

**I recommend publication with major revisions, including a thorough review for improving language.**

*Reviewer #1 had a similar remark. Therefore, while preparing the replies to the reviewers, we submitted the manuscript to an independent English-speaking proofreader to improve the English after taking into account your own corrections.*

**MAJOR COMMENTS**

**My main comment relates to the specificity of the study as a synthetic experiment and with how some of the simplifications/assumptions are presented. I recommend further discussion on the impact of these on a study using real data, and on the estimates of uncertainties which are crucial to any assimilation experiment.**

*Thank you for this important remark. Reviewer #1 raised similar issues and also asked for more discussions regarding the impact of the OSSE's assumptions going towards more realistic experiments. Therefore, the "Discussions" section will be profoundly updated to take into account all the points raised by the reviewers. Please find below our more detailed reply.*

**These are specifically:**

**M#1 - The assimilation of depth rather than elevation**

*We agree that SWOT will provide water surface elevations and not water depths. The assimilation of water depths in the first experiment serves as a reference or benchmark, as we expect it to perform well. Then, going towards more realistic experiments and directly use water elevations as observations, the first challenge would be to deal with the change of reference between the water depths simulated by the model and the water elevations measured by the satellites. To do so, we need to know the elevation of the bottom of the river bed referenced to the same geoid as the observations. Yet, this information is generally not well known. Therefore, we chose to use water surface elevation anomalies as observations because this type of variable does not depend on a reference or a bathymetry.*

*This essential point (use elevation anomalies to overcome the issue of unknown bathymetry) constitutes one of the main motivation for this study. Still, we realized it was not clearly stated neither in the abstract nor the introduction. Therefore, we modified these two sections.*

- *Modified abstract: P.1 L.7-10 + P.1 L.12 + P.1 L.17*

- *Modified introduction: P.4 L.6 + P.4 L.7-8 + P.4 L.10-11*

**M#2 - The fact that the truth is generated by the same model:**
**- no model structure error**
**- there is a "real" Manning to converge to, which might not be the case with real data**
**- even in PE3, the bathymetry is not significantly changed, only the river bed elevation.**

*To our knowledge, the model structure error is still a challenging error to estimate and most of data assimilation studies assume no model structure error. However, when using ensemble-based model, a possibility to deal with this error is to enrich the background ensemble by considering more uncertainties from variables that are not necessarily included in the control vector (including errors in forcings or parameters from both the LSM and the RRM). The capacity of such ensemble to deal with the model structure error can then be tested in a framework where we still use synthetic observations but generated from a different model. We added those remarks to the Discussions.*

*We agree that bathymetry errors are more complex than just errors on river bankful depth Errors on widths and also representativeness errors due to the use of a simplified bathymetry are also important. With the presence of such errors, one could expect to generate updated Manning coefficient distribution with unrealistic values that translate the incoherence between the simulated and observed bathymetry. We added a paragraph in the Discussions dedicated to these errors.*

- *Modified Discussions: P.19 L.12-13, P.19 L.19-20, P.20 L.3-11*

**M#3 - Assumption of perfect forcing**

*Reviewer #1 had a very similar remark. We are aware that the "perfect" forcing assumption is a very strong simplifying assumption as the forcing control the amount of water entering the system. We do not plan on correcting the forcing with data assimilation here but rather calibrating the model given the current forcing. However, to smooth the effect of forcing errors on the assimilation, it is possible to include them in the generation of the background ensemble. A dedicated paragraph in the Discussions section has been added to the manuscript to discuss these aspects.*

- *Modified Discussions: P.19 L.21-29*

**MINOR COMMENTS**

**m#1 - P6, l.21: this is brought up later when the anomalies are assimilated, but more focus should be placed here on the fact that depth is assimilated while SWOT will produce elevations. The conversion from level to depth is one of the big issues with using altimetry in hydrological studies. Does the SWOT simulator directly produce depths? You are assuming known river bed elevation, and this should be clearly specified. (I can see this is mentioned p11, l.26 but that is too late in the paper).**

*You are right. This aspect is not clearly set when describing the observations variables. We modified the corresponding paragraph accordingly.*

- *Modified manuscript in section 3.2.1: P.7 L.16-26*

In the present study, the observed variables are water depths issued from a simplified SWOT simulator. Note that this simulator will produce water depths while the real SWOT satellite will provide water elevations. As in Biancamaria et al. (2011) and Pedinotti et al. (2014), our SWOT simulator replicates SWOT spatio-temporal coverage. At a given date, the simulator selects the ISBA-CTRIP cells contained (at least 50% of their area) in the SWOT ground tracks. Figure 2 shows some selected ISBA-CTRIP cells under the real swaths over the Amazon basin. The true run is used as a basis to get the true water depths $Y_{tk}$. Then, to generate the observation vector $y_{ok}$ from the extracted true water depths, each of them is randomly perturbed by adding a white noise characterized by a standard deviation $\sigma_o$ such that:

Equation (6)

Using water depths observation is a strong simplification of the real SWOT product. Therefore, in order to take into account that SWOT will provide water elevations and not directly water depths, this study will look at the assimilation of both water depths and water anomalies. The method to generate the anomalies will be further detailed in Section 4.2.

**m#2 - P7, l.9: similar issue as previous comment, elevation and not depth is required at 10cm accuracy. The depth accuracy will be much lower. You assume no representativeness error due to scale, but how about level vs. depth?**

*It is true that the 10cm vertical accuracy corresponds to water elevations measurements. We should expect higher errors on water depths, as we do not know the bathymetry. Still, it is complex to quantify such bathymetry errors and they only apply to water depths assimilation. As water depth assimilation is considered as a benchmark, we are still considering 10 cm vertical accuracy in our study for water depth. For water elevation anomalies, the error might even be lower, but as we are only considering white noise, we keep 10 cm for anomalies. We added a few remarks to acknowledge this aspect in the manuscript.*

- *Modified manuscript in section 3.2.1: P.8 L.5-10*

The observation error is the addition of the measurement error and the representativeness error. The first is associated to inherent instrumental errors when processes are observed and the second represents the error introduced when the observed and simulated variables are not exactly the same (in nature or scale). Following the SWOT uncertainty requirements (Esteban Fernandez, 2017), SWOT-like water surface elevation measurements have a vertical accuracy of 10 cm (when averaged over a water area of 1 km2). This uncertainty accounts for measurement errors due to the remotely-sensed acquisition such as instrumental thermal noise, speckle, troposphere and ionosphere effects. Moreover, we omit error correlations along the swath so that observation errors follow a white noise model. Accounting for spatially-correlated observation errors is an active research area in the field of data assimilation (Guillet et al., 2018) that is beyond the scope of demonstrating the feasibility of assimilating SWOT-type data.  In the framework of OSSE, observed and simulated water depths have the same scale as the ISBA-CTRIP model is used to generate both. Therefore, in the following, we assume there is no representativeness error related to scale in the system. However, it is worth acknowledging that we should expect higher errors on water depths, compared to water elevations, as we do not know the bathymetry. Assimilation of water depths is performed as a benchmark, against which assimilation of water anomalies will be compared to. Ultimately, σo is chosen equal to 10 cm for all observed variables (i.e. both water depths and water elevation anomalies).

**m#3 - P8, l11: Would this be necessary if you had a better representation of your measurement error (re: previous comments)? What is the magnitude of this additional error?**

*Not necessarily. In our study, we are using a stochastic version of the EnKF method that is, each member of the background ensemble is updated using the analysis equation 12 (in opposition to deterministic version like the Ensemble Transform Kalman Filter, where the observations randomization is not necessary as the analysis equation is only applied to the ensemble mean and the covariance matrix is estimated from a transformation of the prior covariance matrix). According to Burgers et al. (1998), the randomization of the observation vector should be used so that all the variables within the EnKF are random. Without this, the observation vector is deterministic, which conflicts with the EnKF analysis scheme where the ensemble covariance is used instead of the error covariance. Moreover, we know that the stochastic EnKF tends to under-estimate the analysis error. Therefore, we generate different realization of the observation vector so that the analysis ensemble retains enough variability from one assimilation cycle to another and avoid "ensemble collapse" (when all ensemble members are identical).*

*This randomization is specific to the algorithm itself, not the data. Still, the magnitude of the randomization is generally fixed by the observation error (hence, the randomization will be minimal if the observations are very accurate). Therefore, the observation randomization is done by adding to each observation a random perturbation following a white noise with a standard deviation $\sigma^o$=10cm.This is now specified in the manuscript.*

- *Modified manuscript in section 3.3: P.9 L.18-20*

To avoid ensemble collapse, the observation vector in Eq. (5) is randomized by adding a supplementary white noise with the same observation error standard deviation $\sigma^o$=10 cm (Burgers et al., 1998) such that

$$\forall\, j = 1 \dots n_y, \forall\, l = 1 \dots n_e, y_{k,j}^{o,[l]} = y_{k,j}^o + \epsilon_j^{o,[l]}, \epsilon_j^{o,[l]} \sim N(0, \sigma^o).$$

An observation ensemble is generated

$$Y_{e,k}^o = \left[ y_k^{o,[1]}, y_k^{o,[2]}, \dots, y_k^{o,[n_e]} \right].$$

**m#4 - At some points vague language is used (f.ex: "without really converging"), please be more precise.**

*Thank you for this advice. We carefully went through the manuscript and focused on correcting this language.*

**m#5 - P7, l.25: It is not clearly explained if xk are the Mannings coefficients themselves or the multiplying factors as described above. This issue is repeated elsewhere in the paper and while I understand xk does refer to the factors, please make sure this is clear throughout.**

*Reviewer #1 had a similar comment. Therefore, we modified the manuscript to clarify this aspect.*

- *Modified manuscript in section 3.2.2: P.8 L.21-24*

**m#6 - Figure 8: the black lines cannot really be seen, perhaps increase the width?**

*Thank you for this suggestion. We modified Figures 8 and 10 such that the different curves can be seen more easily.*

**ADDITIONAL COMMENTS FROM SUPPLEMENT**

**m#7 – P2, l.9: it is not unavoidable that reductions in structure uncertainty are linked to increase in parameterization.**

*Following the reviewer's recommendation, we re-wrote this sentence.*

**m#8 – P2, l.27-28: This sentence feels odd right after the mention of important vertical errors.**

*Yes, this is true. In the previous sentence, we implies that the signal with high vertical error was then ignored, diminishing more the spatio-temporal coverage of the observations. To be clearer, the part "or with important vertical error" was deleted.*

**m#9 – P5, l.3: in time I agree, but is it not usually spatially variable?**

*To our knowledge, there are models where a spatially-constant Manning coefficient is applied such as Biancamaria et al (2009), Beighley et al.,(2009) and other with a spatially-distributed Manning coefficients such as Decharme et al. (2012). However, it is true that the most recent model developments use a spatially-distributed coefficient. Therefore, this sentence was added to the manuscript.*

**m#10 – P10, l.10: this is confusing, is this study a standalone PE? This paragraph is unnecessarily complicated.**

*You are right; the term "standalone" is confusing. We withdrew it from the manuscript.*

**m#11 – P11, l.26: this should be mentioned way earlier in the text.**

*Please see reply to m#1.*

**m#12 – P12, l.8: isn't it river bed elevations you are talking about?**

*Yes, this is right. We changed the term in the manuscript.*

**m#13 – P15, l.1-5: I don't see a clear distinction between the behavior of these 2 groups in the figures**

*This is true. Besides, for the interpretation of the results in the next paragraph, zones 1 and 9 are interpreted along with zones 2, 3, 4 and 5. Therefore, this item is deleted and zones 1 and 9 are now cited in the first item.*

**m#14 – P16, l.1-2: It is overstating it that the bathymetries are different, the shape itself remains the same, only the bed elevation is shifted.**

*This is true. We re-adjusted this sentence in the corrected manuscript and worked on the paragraph.*

- *Modified manuscript in section 6.2: P.17 L.22-23:*

In the PE2 experiment, there is no difference of bathymetry between the simulated and observed water anomalies while the river bankful depth is different in the PE3 experiment.

**m#15 – p17, l.2-6: I find it odd that this is how you begin the discussion. If that is why assimilation is introduced, then the natural conclusion would be to update the river bed elevation.**

*It is true that this sentence was confusing. This part has been rewritten.*

- *Modified discussions P.18 L.24-26 + P.19 L.6-8*

**m#16 – P17, l.14-15: This will help only if width, bank slope etc are also correct or corrected**

*You are right. Following previous reviewers' comments, this point was covered elsewhere in the Discussions and the current sentence does not appear anymore in the manuscript.*

***Additionnal references***

*Sylvain Biancamaria, Paul Bates, Aaron Boone, Nelly Mognard. Large-scale coupled hydrologic andhydraulic modelling of the Ob river in Siberia. Journal of Hydrology, Elsevier, 2009, 379 (1-2), pp.136-150. doi 10.1016/j.jhydrol.2009.09.054*

Beighley, R. E., Eggert, K. G., Dunne, T., He, Y., Gummadi, V.,and Verdin, K. L.: Simulating hydrologic and hydraulic processes throughout the Amazon River Basin, Hydrol. Process., 23, 1221–1235, doi:10.1002/hyp.7252, 2009

Hess-2019-242-RC3
Reply to Paul Bates' comments

**General Comments**

**In addition to the comments of the previous referees, I would add that I think the paper also needs to better articulate what is the new contribution to knowledge made by the work. Previous papers have established that SWOT data is very likely going to allow us to better constrain parameters and states in a variety of environmental models, but I was missing what in addition to this that we learn from the present work. There are a number of specific technical differences here to the previous papers by Pedinotti et al (2014) and Emery et al (2016) which are described in the last paragraph of the introduction, but I am not sure these changes lead to very different conclusions.**

*The authors would like to thank referee Paul Bates for his helpful comments. Please find below our replies with which we hope to address his concerns. To easily address all comments, we numbered each comment: they are referenced with a "M" for the major comments and with a "m" for the minor comments.*

*The abstract, the introduction (the latest paragraphs) and the discussions sections have been heavily modified to take into account several remarks from the three reviewers. Therefore, we refer the reviewers directly to the corrected annotated manuscript. In the annotated manuscript, we used a color code to differentiate which reviewer made the comment and suggested a modification: comments from reviewer 1 (Hessel Winsemius), 2 (Claire Michailovsky) and 3 (Paul Bates) are in purple, orange and green respectively. Each modification is also referenced by a code in bracket indicating the reviewer (R#***) and the type and index of the comment (M/m#***) such as: "[R#3-M#1]".*

**M#1 - Overall the abstract states that the assimilation scheme "is able to retrieve the true value of the Manning coefficients within one assimilation cycle most of the time", but isn't this ability already well established, at least in general terms? What do we learn about the likely potential of the SWOT data that we did not know before? Alternatively, what do we learn about the kinds of assimilation schemes that are going to work well with the SWOT data? In addressing this issue the comments of the other referees about the rather specific nature of the numerical experiment are pertinent: if the experiment does not reflect the assimilation problem that will be faced in practice when ISBA-CTRIP is confronted with SWOT data then the conclusions drawn may not be easy to generalize.**

**I think a significant effort needs to be made to re-draft the abstract introduction to more clearly identify what is the new contribution to knowledge made by the paper. I'm sure there is one, so this is more a question of properly assembling the arguments in order to bring this out.**

*Beyond the objective to show the capability of assimilating SWOT-like data to correct the model's parameters, we wanted to introduce the assimilation of water surface anomalies as observed variables. Indeed, hydrological models usually produce water depths referenced to the bottom of the modeled bathymetry while SWOT will provide water surface elevations referenced to a reference surface (a geoid or an ellipsoid). Then, one of the biggest challenges in assimilating satellite-based water surface elevations remain in how they can be compared to simulations without introducing a large bias in the system. Therefore, we propose a method where we avoid the comparison between the model and observations reference by using the water surface anomalies that can be both*

*extracted from the model or the observations and compared. The use of anomalies offers then a method to handle SWOT data when we do not have any information on the bathymetry. However, it was required to show that assimilating water elevation anomalies would provide comparable results than assimilating water depths. This is the purpose of our study.*

*By gathering the comments from all reviewers, we realized this aspect was not enough emphasized in our manuscript. Therefore, we modified the abstract and the introduced to better assert this aspect. Moreover, the water surface anomalies are also introduced earlier in the manuscript: they were initially presented in Section 4.2 describing the assimilation strategy and they are now mentioned in Section 3.2.1 with the description of the observation variable. Finally, following a comment from reviewer #2, the manuscript has been cleaned to avoid the confusion between water depths and water surface elevations.*

*Our methodology was also modified in light of Pedinotti et al. (2014)'s results. We are now using a more complex assimilation scheme, the asynchronous EnKF over a 21-day assimilation window and we adopted a zonal approach to correct the Manning coefficient distribution.*

*Moreover, we are aware of limitations of our experiments. Going towards more realistic experiments will require some modifications to our OSSE. For example, more realistic model errors are needed, such more complex bathymetry errors, forcing and LSM parameters errors, model structure errors. More realistic SWOT observation errors must also be taken into account in future works. In our current study, we partially look at errors on the riverbed bankful depth. The other types of uncertainties are of a different nature and would require more processing than just the use of anomalies as observations. As this represents a first test with water surface anomalies, we limited our experiments to the proof-of-concept*

*and the test of the river bankful depth error. However, following the reviewers' recommendations, we intensively reworked the Discussions section to discuss the other types of uncertainties: how they would affect the assimilation performances and how they can be handled.*

- *Modified abstract, P.1 L.9-10, P.1 L.18-19*

- *Modified introduction, P.4 L.8-10*

**List of all relevant changes to the manuscript**
**(following the annotated manuscript outlines)**

- **P.1, L.4 [R#1-M#1]:** added the lifetime of SWOT

- **P.1, L.7-10 [R#2-M#1]+[R#3-M#1]:** additionnal remarks to better contextualize the study and its scientific contribution, i.e. the assimilation of water surface anomalies

- **P.1, L.12 + P.1, L. 17 [R#2-M#1]:** additional remarks to clearly state that the study will look at the assimilation of water surface anomalies

- **P.2, L.10, L.14 [R#1-m#1]:** rewording

- **P.2, L.16 [R#2-m#7]:** rewording

- **P.1, L.19 [R#3-M#1]:** additional remark to show the contribution of the study

- **P.3, L. 29 – P.4, L.4 [R#1-M#1]:** additional remark to better contextualize the study and justify the chosen parameter estimation framewok

- **P.4, L.7-8 [R#2-m#11]:** additionnal remark to clearly differentiate simulated water depths from observed water surface elevations and how it impacts the study

- **P.4, L.8-10 [R#3-M#1]:** additional remark to better define the study's objectives

- **P.4, L.10-11 [R#2-M#1]:** additional remark to better contextualize the study

- **P.4, L.12-14 [R#1-M#1]:** additional remark to better contextualize the study and justify the chosen parameter estimation framewok

- **P.4, L.22-23 [R#1-m#2]:** rewording to better explain the different variable scales

- **P.5, L.8 [R#1-m#3]:** additional definition

- **P.5, L.28 [R#1-m#6]:** corrected typing error

- **P.5, L.29-30 [R#2-m#8]:** modified statement on the use of the Manning coefficient in global scale hydrologic model

- **P.6, L.3-6 [R#1-m#7]+[R#1-m#8]:** rewording and corrected typing error to better explain Equation 1

- **P.6, L.7-8 [R#1-m#9]:** rewording to better define the velocity variable

- **P.6, L.10 [R#1-m#10]:** added missing definition of river storage S

- **P.7, L.16-17 [R#2-m#1]:** additional remark to clearly differentiate simulated water depths from observed water surface elevations

- **P.7, L.24-26 [R#2-m#1]:** additional remark to introduce water anomalies

- **P.8, L.5-10 [R#2-m#2]:** additional remark to discuss the observation error model

- **P.8, L.18-24 [R#1-M#2]:** additional paragraph to better justify the choice of control variables

- **P.9, L.18-20 [R#2-m#3]:** rewording to better present the observation randomization process

- **P.11, L.14-17 [R#1-m#11]:** additional remark on OSSE's assumptions

- **P.13, L.8-10 [R#1-m#14]:** additional remarks on what we expect the assimilation results to be

- **P.13, L.26-30 [R#1-m#14]:** additional remarks on what we expect the assimilation results to be

- **P.14, L.26-28 [R#1-m#15]:** additionnal remark on the physical meaning of the spinup time

- **P.16, L.12-13 [R#1-M#2]:** additional remark to better explain how the control variables are defined

- **P.16, L.19-20 [R#1-M#1]:** additional remark to introduce a new figure in appendix

- **P.17, L.12-15 [R#1-m#17]:** additionnal remark on the physical meaning of the assimilation results

- **P.17, L.22-23 [R#2-m#14]:** rewording

- **P.17, L.25-26 [R#1-M#1]:** additional remark to introduce new tables in appendix

- **P.18, L.4-9 [R#2-m#4]:** rewording

- **P.18, L.24-26 [R#2-m#15]:** rewording

- **P.18, L.29-30 [R#1-m#19]:** rewording

- **P.19, L.6-8 [R#2-m#15]:** rewording

- **P.19, L.12-13 [R#2-M#2]:** rewording

- **P.19, L.19-20 [R#2-M#2]:** additionnal discussions on the impact of bathymetry error on the corrected control variables

- **P.19, L.21-29 [R#1-M#1]+[R#1-m#18]:** additionnal paragraph on the impact of forcing and LSM errors

- **P.20, L.3-11 [R#2-M#2]:** additionnal paragraph on the impact of model structure error

- **P.20, L.17-27 [R#1-m#12]:** additionnal paragraph on more realistic observation error model

- **P.21, L.31-32 [R#1-M#1]:** added more study perspectives

- **P.21, L.35 – P.22, L.1-3 [R#1-M#1]:** additionnal perspectives on dual state-parameter estimation perspective

- **P.23, L.2-5:** new appendix in which the table explaining the spinup sensitivity test is moved to unload the main manuscript.

- **P.23, L.6-7 [R#1-m#16]:** new appendix with additionnal figure

- **P.23, L.8-16 [R#1-M#1]:** new appendix with additional tables showing the assimilation performances in terms of water depth and discharge RMSEn

[revised manuscript text omitted]

---

## Author Response (AR2)

Hess-2019-242

Letter to the editor

The authors would like to thank editor Marcus Hrachowitz for its very positive return on our manuscript, now entitled « Assimilation of wide-swath altimetry water elevation anomalies to correct large-scale river routing model parameters ». Following his advice, we modified the title of the manuscript.

Kind regards
Charlotte Emery and co-authors.